# Synaptic ribbon dynamics after noise exposure in the hearing cochlea
Noura Ismail Mohamad [1,2], Peu Santra[1,2], Yesai Park[1], Ian R. Matthews[1], Emily Taketa [1] &
Dylan K. Chan [1] ✉

Moderate noise exposure induces cochlear synaptopathy, the loss of afferent ribbon synapses between cochlear hair cells and spiral ganglion neurons, which is associated with functional hearing decline. Prior studies have demonstrated noise-induced changes in the distribution and number of synaptic components, but the dynamic changes that occur after noise exposure have not been directly visualized. Here, we describe a live imaging model using RIBEYE-tagRFP to enable direct observation of pre-synaptic ribbons in mature hearing mouse cochleae after synaptopathic noise exposure. Ribbon number does not change, but noise induces an increase in ribbon volume as well as movement suggesting unanchoring from synaptic tethers. A subgroup of basal ribbons displays concerted motion towards the cochlear nucleus with subsequent migration back to the cell membrane after noise cessation. Understanding the immediate dynamics of synaptic damage after noise exposure may facilitate identification of specific target pathways to treat cochlear synaptopathy.

Noise-induced hearing loss (NIHL) typically involves primary damage to sensory hair cells followed by secondary degeneration of connected auditory neurons. However, moderate levels of noise exposure can result in degradation of afferent synapses between inner hair cells (IHCs) and spiral ganglion neurons (SGNs)[1,2]. This noise-induced cochlear synaptopathy (NICS) leads to the permanent severing of many surviving IHCs from their sensory innervation[3]. While hair-cell loss is captured by increases in auditory thresholds—the standard clinical assay of hearing loss—loss of synapses is not, leading to "hidden" hearing loss (HHL). HHL has been associated with characteristic changes in auditory brainstem response (ABR) waveforms[1,4]. Behaviorally, HHL is thought to be associated with significant social, psychological, and cognitive implications, including perceptual difficulties in complex acoustic environments[5], tinnitus[6–8], and hyperacusis[6].

The presynaptic region of IHC-SGN synapses is characterized by the presence of electron-dense structures implicated in the modulation of trafficking and fusion of glutamatergic synaptic vesicles. These presynaptic ribbons are apposed by postsynaptic NMDA, kainate, and AMPA-type receptors on peripheral afferent fibers[9–11]. In the cochlea, release and reuptake of the primary neurotransmitter glutamate is tightly regulated to optimize fast synaptic transmission[12–14]. The mechanism of NICS is not fully elucidated, but is thought to primarily stem from glutamate excitotoxicity — increased release of glutamate from overstimulated IHCs[15–18]. Noise exposure, anoxia, and treatment with ionotropic glutamate receptor agonists

(such as kainic acid (KA)) cause excitotoxic damage to afferent synapses, prompting a massive influx of water and ions, such as calcium, into auditory fibers[1,11,19–21]. This can lead to swelling, stimulation of proteolytic enzymes, and an increase of toxic reactive oxygen and nitrogen compounds, which can result in neuronal degeneration and a permanent loss of function[22].

Much of what is known on the mechanisms behind NICS in IHC ribbon synapses stems from immunohistochemical studies, by staining against protein markers on pre-synaptic ribbons (anti-c-terminal binding protein 2 (anti-CtBP2)), post-synaptic glutamate receptors (anti-GluR), and SGNs (anti-NF) in fixed cochlear tissue. The reported changes in synapses after acoustic overstimulation or ototoxic drug treatments include the loss or dysfunction of GluR and postsynaptic density protein 95 on type I afferent terminals[23,24], the dissociation or migration of pre-synaptic terminals along the basolateral membrane of IHCs[1,25], and terminal swelling[15,26]. The damage to synapses caused by noise exposure also appears to worsen the decline of SGNs that occurs with age[19,24].

Traditional immunohistochemical staining methods do not reveal real-time dynamics of ribbon synapses under physiologic or pathophysiologic stimuli. Live imaging of perturbed synapses helps fill in these gaps in knowledge. Work in goldfish retina bipolar cells in which a short peptide with affinity for the CtBP subunit of the RIBEYE protein – the major component of ribbons — or, later, calcium indicators, were introduced via whole-cell patch pipette, provided critical insight into the development and organization of synaptic ribbons[27,28]. Subsequent development of a mouse

[1]Department of Otolaryngology-Head and Neck Surgery, University of California, San Francisco, San Francisco, CA, USA. [2]These authors contributed equally: Noura Ismail Mohamad, Peu Santra. ✉e-mail: dylan.chan@ucsf.edu

transgenic line in which RIBEYE is tagged by a red fluorescent protein (RIBEYE-tagRFP), enabled visualization of the dynamic assembly of ribbon synapses during synaptogenesis and maturation in the mouse retina[29]. This study showed that the endogenous, genetically encoded RIBEYE-tagRFP signal is an effective tool for tracking the development of synaptic ribbons, and that while synaptic ribbons can affect the stability of nascent bipolar cell synapses, they are not essential for establishing these synapses in vivo.

Similarly, in RIBEYE knockout mice lacking synaptic ribbons in hair cells, only minor disruptions in the function of hair-cell synapses and minor auditory impairments are present, suggesting the existence of compensatory mechanisms in this model system[30–32]. However, RIBEYE was shown to play a crucial role in organizing presynaptic Cav1.3 calcium channels, although the localization of these channels at the hair-cell synapse remained unaffected. In the absence of RIBEYE, numerous small Cav1.3 clusters were observed at each synapse, deviating from the usual single organized structure[30,32].

Initial imaging studies of live ribbon synapse activity predominantly relied on dissociated retinal cells to explore structural ribbon dynamics. RIBEYE-tagEGFP transgenic zebrafish have been used to investigate hair cell ribbons in the lateral line system over extended periods without compromising the structural integrity, osmolarity, or mechanical forces surrounding the cells – factors that could influence calcium influx and synaptic activity[33]. Discrepancies with previous studies highlight the potential for advanced imaging technologies to provide deeper insights into the dynamic nature of ribbon synapses and vesicle interactions in different cell types and experimental conditions.

In the present study, we describe the use of a live synaptic imaging model to observe the behavior of pre-synaptic ribbons in an organotypic tissue culture model of neonatal RIBEYE-tagRFP mouse cochleae, as well as in 7–10 week-old RIBEYE-tagRFP mature hearing mouse cochleae. Excitotoxic lesioning was induced by KA for neonatal mice, and by noise exposure for mature, hearing mice. We tracked presynaptic ribbon dynamics to visualize change over time and investigate the mechanism of glutamate excitotoxicity and noise exposure in live cells, enabling direct observation of the initial events underlying NICS.

## Results

### Endogenous RIBEYE-tagRFP signal accurately represents pre-synaptic ribbons in cochlear inner hair cells

The RIBEYE-tagRFP mouse has previously been shown to accurately label pre-synaptic ribbons in the retina[29]. We sought to validate this model in cochlear IHCs from P5 neonatal explant cultures. We first performed immunohistochemistry with antibodies against Neurofilament, CtBP2, and GluR2 to label neurons, pre-synaptic ribbons, and post-synaptic receptors, respectively. Merged images of RIBEYE-tagRFP and anti-NF signals (Fig. 1a) demonstrate that SGNs and IHC-SGN connections have been structurally preserved in our in vitro culture model. The IHC-SGN synapses were further confirmed to be intact through visualization of adjacent RIBEYE-tagRFP and anti-GluR2 labeling, indicating paired synapses (Fig. 1b). Anti-CtBP2 immunohistochemical labeling and endogenous RIBEYE-tagRFP signal overlapped, both qualitatively (Fig. 1c) and quantitatively: the number of puncta per IHC (Fig. 1d) and the mean volume of puncta (Fig. 1e) identified using RIBEYE-tagRFP signal versus anti-CtBP2 signal alone were not statistically significantly different, and a cytofluorogram scatter plot shows a strong positive correlation (Pearson's coefficient: 0.75) between RIBEYE-tagRFP and anti-CtBP2 signals (Fig. 1f). Taken together, these findings demonstrate that the endogenous, genetically encoded RIBEYE-tagRFP signal accurately represents gold-standard fixed immunohistochemical staining and is an appropriate tool for live imaging of pre-synaptic ribbons in cochlear IHCs.

### Dynamic three-dimensional imaging and tracking of presynaptic ribbons in cochlear inner hair cells

We next sought to establish a method for live imaging, identification, and tracking of RIBEYE-tagRFP puncta, as markers of pre-synaptic ribbons

(Fig. 2). Cochlear hair cells were loaded with FM1-43, a live dye that enters through functioning mechanoelectrical transduction channels[34]. The FM1-43 signal enabled establishment of a stable three-dimensional reference frame for determination of RIBEYE-tagRFP punctal position over time. Serial confocal z-stacks were acquired encompassing the entire IHC region within the optical field, and the entire volume of FM1-43 hair cells was rendered in three dimensions (Fig. 2a). To account for spatial drift over imaging time, the FM1-43 signal was taken as a frame of reference in the three-dimensional reconstruction of the presynaptic ribbons. The stability of this reference frame was ascertained in each experimental condition by measuring the speed with which the center of mass of the reference frame moved; this speed did not vary with experimental perturbation (Supplementary Fig. S1). With this technique, we were able to track presynaptic ribbons in three dimensions in multiple IHCs over time, allowing for visualization and time-lapse observation of dynamic ribbon parameters such as speed, volume, displacement, and directionality (Fig. 2b–d). Acquisition of one complete confocal Z-stack required 5 min on average; for serial measurement, therefore, the average minimum time interval between successive images was 5 min, which determined the temporal resolution for dynamic measurements.

### Kainic acid treatment induces movement of ribbons in neonatal cochlear cultures

Exposure to KA to induce glutamate excitotoxicity is a well-described model of cochlear synaptopathy used in neonatal cochlear explant cultures[35–37]. We first sought to validate this model in our system. 0.4 mM KA, or vehicle, was applied to RIBEYE-tagRFP neonatal cochlear cultures for 2 h to mimic glutamate excitotoxicity, and cochleae harvested 24 h later for anti-GluR2 immunohistochemistry and visualization of tagRFP signal. The number of ribbon puncta remained unchanged from the control, but the number of post-synaptic puncta was reduced, making the number of paired synapses reduced following KA treatment consistent with previous studies[35–37] (Supplementary Fig. S2). We then performed live imaging of RIBEYE-tagRFP neonatal cochleae and obtained serial confocal z-stacks of IHCs before and 24 h after 2-h application of 0.4 mM KA. Control cochleae exhibited a decrease in the number of RIBEYE-tagRFP puncta after 24 h in culture. Cochleae treated with KA also showed a decrease in puncta number; the number of puncta, however, was not statistically significantly different between control and KA-treated cochleae at any timepoint (Fig. 3a). This is consistent with prior findings that the number of pre-synaptic ribbons is not affected by glutamate excitotoxicity within 24 h[25,38]. Ribbon volume was also unchanged in control and KA-treated cochleae 24 h after drug exposure (Fig. 3b). We then assessed movement of synaptic ribbons by measuring the difference in position between a pair of images and dividing this displacement by the time interval between the paired images. Comparison of speed measurements in live-imaged cochleae and the same cochleae after fixation confirmed that the live-imaged puncta had greater movement, thus excluding imaging or mechanical artifact (Supplementary Fig. S3). The speed of RIBEYE-tagRFP puncta was not different prior to treatment in control and KA cochleae, and increased in both groups 24 h after treatment.; however, KA-treated cochleae had significantly greater increase in speed compared with control cochlea (Fig. 3c).

These speed measurements were performed using a single pair of images before and 24 h after a 2-h KA treatment. We then performed serial timecourse imaging during KA exposure. The increase in speed was present immediately after KA exposure (Fig. 3d) and remained elevated over a 1-hr recording period (Fig. 3e). Though the speed with which the ribbons moved was increased, there was no consistent directionality to ribbon movement. The 95% confidence interval of the mean velocities for overall movement as well as in each of the relevant cochlear axes spanned the origin, suggesting that the entire population of ribbons, taken as a whole, does not exhibit any significant directionality in movement. Indeed, the velocity vectors corresponding to punctal movement along the apical-basal and modiolar-pillar axes were evenly distributed in both control and KA-treated cochleae (Fig. 3f, g).

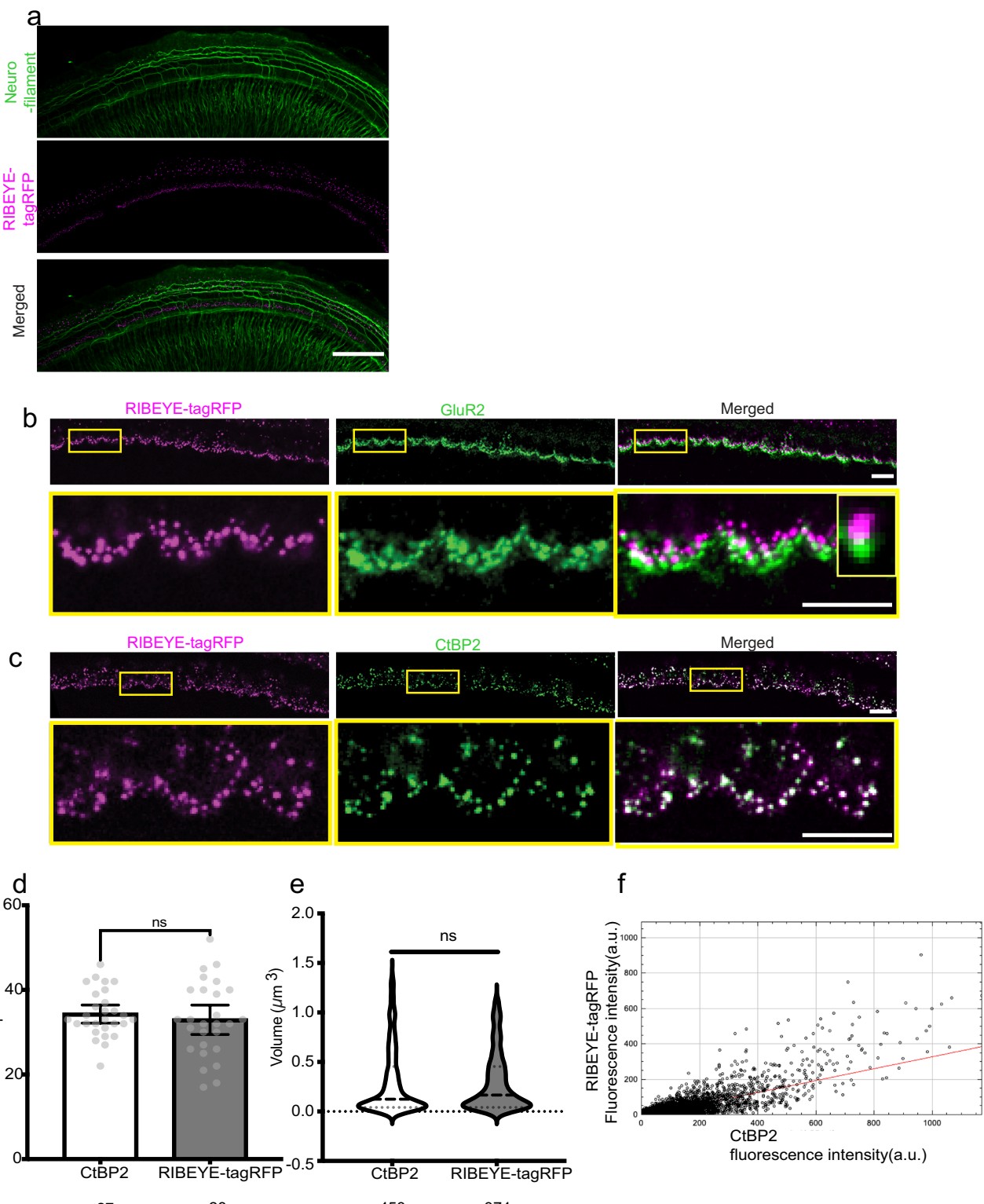

**Fig. 1 | Validation of RIBEYE-tagRFP imaging in cochlear hair cells.**
**a** Representative images through the mid portion of the middle turn of the cochlear explant hair cell region. Presynaptic ribbons marked by RIBEYE-tagRFP in magenta, found adjacent to SGNs labeled with anti-neurofilament in green. Scale bar = 50 μm. **b** Presynaptic ribbons labeled with adjacent endogenous RIBEYE tag-RFP (magenta, left) and post-synaptic densities with anti-GluR2 (green, middle) demonstrate paired synapses (merged, right). Scale bars = 5 μm Inset: example of a paired synapse. **c** Presynaptic ribbons labeled with endogenous RIBEYE-tagRFP signal (magenta, left) and anti-CtBP2 (green, middle) demonstrate near complete co-localization (white, merged, right). **d** The number of RIBEYE-tagRFP and anti-CtBP2 puncta per IHC was not different ($N$ = 27, 28 IHCs, gray dots). **e** Volume of RIBEYE-tagRFP and anti-CtBP2 puncta was not different ($N$ = 453, 374 puncta). **f** A cytofluorogram scatter plot shows colocalization of RIBEYE-tagRFP and anti-CtBP2 signals. Pearson's co-efficient, r = 0.75. ns not significant.

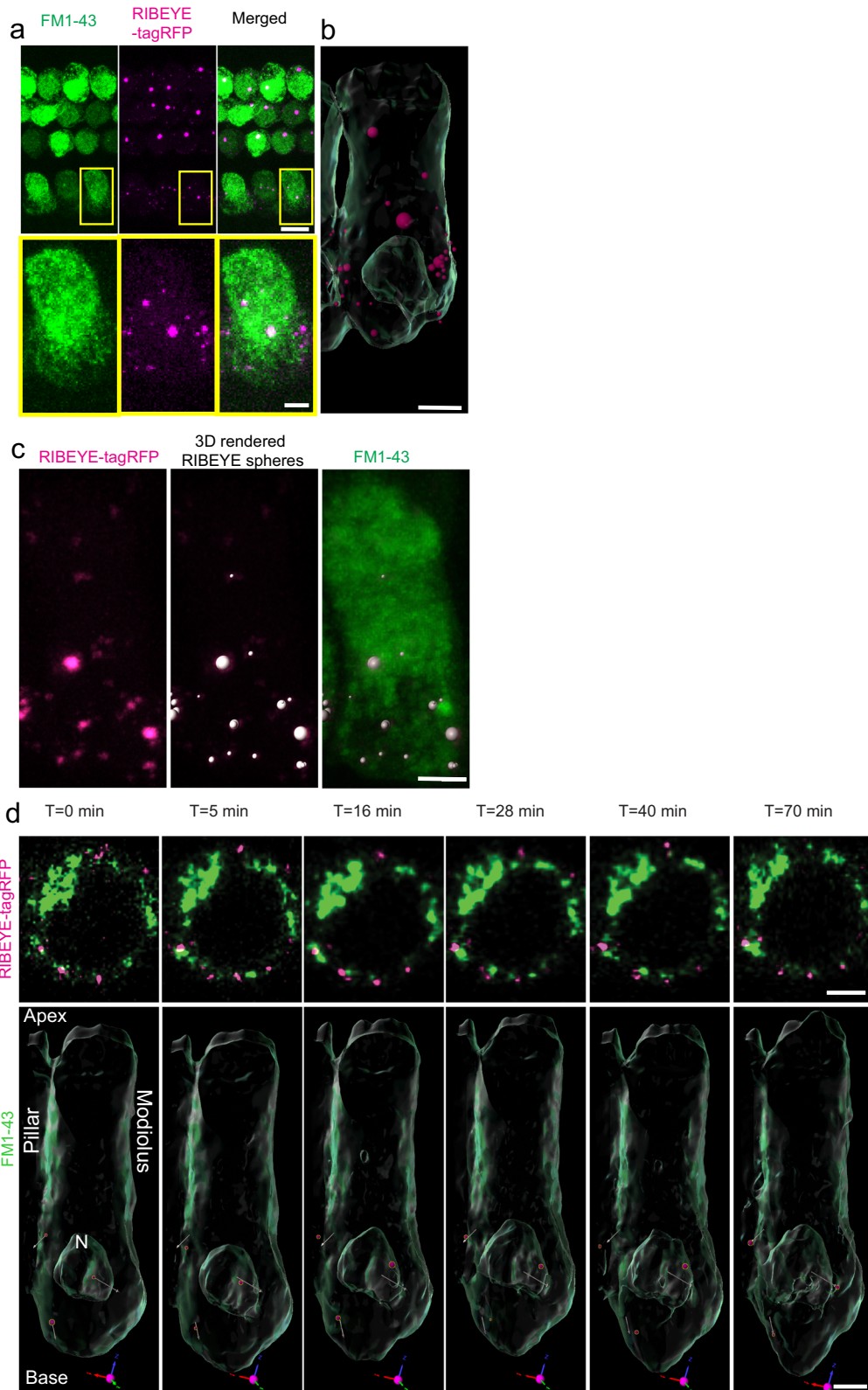

**Fig. 2 | Live imaging of presynaptic ribbons along the basolateral membrane of IHCs in neonatal cochlea. a** IHCs are labeled with FM1-43 dye (green, left) and endogenous RIBEYE-tagRFP signal (magenta, middle) detected simultaneously (merged, right). Scale bar = 5 μm (top panels); 2 μm (bottom panels). **b** The single IHC indicated by the yellow box in A was rendered in 3-D (Imaris; scale bar = 2 μm), with selected RIBEYE-tagRFP puncta rendered as spheroids with volumes equivalent to the detected punctal volume. **c** RIBEYE-tagRFP puncta from a different IHC (magenta, left) are rendered by Imaris as spheroids (center) and superimposed upon the FM1-43 signal (right) for analysis. Scale bar = 1 μm **d** Serial slice montage of 2D and 3D timelapse (upper and lower images, respectively; scale bar = 2 μm) with FM1-43 and RIBEYE-tagRFP signals to track presynaptic ribbon dynamics over time. Arrows in the 3D images represent the overall displacement of the associated puncta from the first to last timepoints, with the length proportional to the distance traveled.

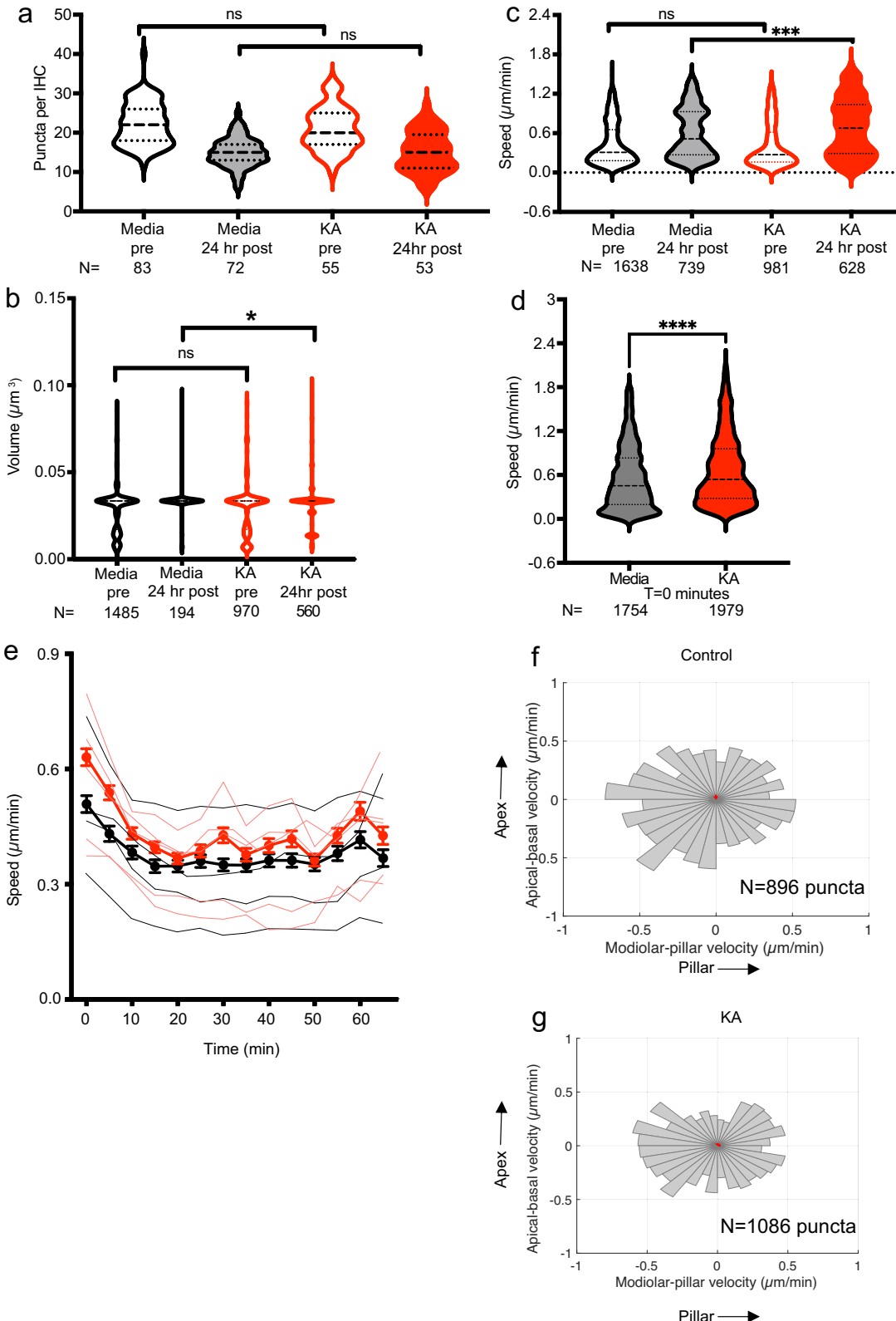

**Fig. 3 | Pre-synaptic ribbon characteristics after excitotoxic lesioning. a** Number of puncta per IHC in live-imaged specimens after 2 h KA vs control, before (pre) and after (24 h post) KA or control (media) treatment. Violin plots indicate median (dashed line) and IQR (dotted lines). Black outlines (control) with white fill (pre) and gray fill (post); red outlines (KA) with white fill (pre) and red fill (post). *N* number of IHCs. Punctal volume (**b**) and speed of punctal movement (**c**) after KA or control (media) treatment. *N* = number of puncta for each condition. **d** Speed was also measured immediately after KA or vehicle treatment in cochlear cultures.
**e** Timecourse of punctal speed for 65 min after KA treatment (red) compared with control (black). Dots and thick lines: Means ± 95% CI; thin lines: individual time-courses. **f, g** Apical-basal velocity vs modiolar-pillar velocity in control or KA treated cochlea. **b–d, f–g**. *N* = number of puncta. **a–d**. *p < 0.05, ***p < 0.001; ****p < 0.0001; ns not significant.

## RIBEYE-tagRFP puncta are less mobile and larger in volume in juvenile compared with neonatal cochleae

Next, we sought to perform live ribbon imaging in juvenile, mature-hearing cochleae. The genetically encoded RIBEYE-tagRFP model provided an opportunity to visualize presynaptic ribbons immediately after euthanasia and cochlear extraction, overcoming the significant difficulty that juvenile cochlear explants rapidly degenerate after extraction, limiting live imaging potential. After rapid dissection and FM1-43 loading, explant cochleae were imaged within 15 min of euthanasia (Fig. 4a). Hair-cell health was monitored using the FM1-43 signal, and cells determined to be intact up through 30 min of recording, after which evidence of bleb formation and extrusion of hair cells was noted. Compared with neonatal cochleae, there were fewer RIBEYE-tagRFP puncta in juvenile inner hair cells (Fig. 4b); however, the punctal volume was significantly higher (Fig. 4c). Both observations are qualitatively consistent with prior reports[39–43]. Additionally, RIBEYE-tagRFP puncta moved at significantly lower speed in juvenile compared to neonatal cochleae (Fig. 4d). Though punctal volume was increased in juvenile cochlea, mean intensity was decreased, suggesting that the puncta become less dense with age (Fig. 4e). These findings are consistent with the known development of ribbon synapses in cochlear inner hair cells, in which ribbons decrease in number, increase in volume, and become more stable as they form mature, paired synapses[39–42].

## RIBEYE-tagRFP puncta are more mobile in noise-exposed juvenile, mature-hearing cochlear cultures compared to unexposed counterparts

Exposure to moderately loud noise, sufficient to cause a temporary shift in hearing thresholds that returns fully to baseline levels, leads to cochlear synaptopathy, in which SGN-IHC synapses are permanently lost. In 7–10 week-old mice, exposure to 8–16 kHz octave-band white noise at 98 dB SPL for 2 h is a well-established model for NICS[44]. The RIBEYE-tagRFP mouse model provides an opportunity to examine the dynamic behavior of cochlear synaptic ribbons after noise exposure. We exposed RIBEYE-tagRFP mice to synaptopathic noise, which caused a temporary threshold shift at 8 and 16 kHz (Supplementary Fig. S4). Unexposed, age-matched littermate mice imaged on the same day were used as controls. Serial live imaging was performed to measure static (number, position, and volume) and dynamic (displacement and speed) characteristics of pre-synaptic ribbons immediately after completion of the 2-h noise exposure (Fig. 5a). There was no significant difference in the number of RIBEYE-tagRFP puncta between control and noise-exposed mice (Fig. 5b). However, puncta from noise-exposed mice demonstrated significantly higher ribbon volumes compared with unexposed mice, consistent with prior reports of noise-induced ribbon swelling[1,45,46] (Fig. 5c). There was a striking difference in ribbon speed, with ribbons from noise-exposed mice moving significantly more rapidly than those from unexposed mice (Fig. 5d).

In a separate experiment, noise-exposed mice were imaged 2 weeks after noise exposure and compared on the same day to control, unexposed, age-matched littermate mice (Fig. 5e). As was observed immediately after noise exposure, the number of pre-synaptic ribbons was not different between noise-exposed and unexposed animals, though variance increased significantly, which may reflect long-term differences between animals in synaptic changes after noise exposure (Fig. 5f). However, there was a large, statistically significant increase in punctal volume (Fig. 5g) and a persistent, small increase in punctal speed (Fig. 5h) in noise-exposed mice 2 weeks after completion of noise exposure.

## Basal-most synaptic ribbons move towards apex after noise exposure

Similar to neonatal cochleae, synaptic ribbons in control as well as noise-exposed juvenile cochleae did not demonstrate any consistent direction of movement overall (Fig. 6A). However, we separately analyzed the 25% of ribbons that were closest to the basal pole of the hair cells. Compared with ribbons found more towards the apical aspect of the IHCs, these basal-most ribbons exhibited consistent movement towards the hair-cell apex and

nucleus, as well as a large increase in volume in noise-exposed cochleae immediately after synaptopathic noise exposure (Fig. 6b, c). This subset of ribbons was examined further over a 20-min recording period. In control cochleae, 127/278 (45.7%) ribbons were able to be followed for the entire recording period; in noise-exposed cochleae, 93/194 (47.9%) persisted, which was not significantly different. There was no difference in subcellular distribution between puncta that persisted or vanished. Both ribbons that vanished and those that persisted demonstrated volume increase and initial movement towards the apex in noise-exposed cochleae (Fig. 6d, e). As the persistent ribbons were followed over time, the initial apical movement in noise-exposed cochleae reversed course, such that towards the end of the recording period, 20 min later, they were moving back towards the base. In contrast, ribbons in control cochlea exhibited no consistent direction of movement at any point (Fig. 6f).

## Discussion

Pre-synaptic ribbons are electron-dense structures in the retina, inner ear and pinealocytes that dock synaptic vesicles, facilitating sustained vesicle release in pre-synaptic active zones. In the cochlea, overexposure to moderate levels of sound leads to cochlear synaptopathy — loss of synaptic connections between sensory IHCs and afferent SGNs — and auditory functional disability. Using a RIBEYE-tagRFP transgenic mouse, we performed live imaging in cochlear explants to visualize the dynamic behavior of pre-synaptic ribbons in live hair cells in real time and three dimensions, and demonstrate that chemical overstimulation and noise exposure induce random, as well as directional movement of subpopulations of pre-synaptic ribbons.

Prior studies in mammalian cochleae have always used immunohistochemistry to examine ribbon synapses in fixed tissue at a single timepoint. Although these studies have revealed immense amounts of knowledge about pre- and post- synaptic structures, they are unable to provide dynamic spatiotemporal information, precluding direct investigation of the effect of noise and other ototoxic insults on ribbon dynamics. In contrast, the genetically encoded model of RFP-tagged RIBEYE allows monitoring of movement of pre-synaptic ribbons in real time in live cochlear inner hair cells.

We identified pre-synaptic ribbons using endogenous RIBEYE-tagRFP signals, while IHCs were delineated using FM1-43; this enabled us to normalize the reference frame around the stable 3D framework of FM1-43 signal and isolate ribbon movement relative to the hair cells. Similar to prior studies of live-cell speed in tumor-associated macrophages[46], we utilized the Spots function in Imaris to identify and render individual RIBEYE puncta. This strategy proved to be ideal for detecting the small RIBEYE puncta and assign a globular-like shape to each. As the first measure during image analysis, we used the Quality filter in Imaris[47]. If the threshold value is too low, erroneous spots are picked; alternatively, if the value is too high only the brightest puncta are picked. Together with Quality, the Intensity Threshold parameter needed to be optimized separately for neonatal and adult imaging, and it was necessary to perform contemporaneous control measurements with age-matched mice to ensure experimental rigor.

In the current study, we performed live imaging in neonatal cultures as well as an explant preparation of the mature, hearing cochlea in juvenile, 7–10-week-old mice. In the case of neonatal cochlear cultures, this imaging enabled investigation of ribbon properties and behavior at multiple timepoints before and after experimental manipulation, including chemical excitotoxicity using KA, in the same specimen. In juvenile, mature-hearing cochlear explants, inability to maintain healthy tissue ex vivo greater than 30 min precluded similar prolonged ex vivo investigation; however, this model was uniquely able to provide direct insight into ribbon behavior after physiologically relevant acoustic noise exposure known to cause cochlear synaptopathy.

We first validated our live imaging and processing model against standard immunohistochemical methods, demonstrating our ability to accurately detect pre-synaptic ribbons in real time. We then evaluated the effect of overstimulation on pre-synaptic ribbons in an established KA

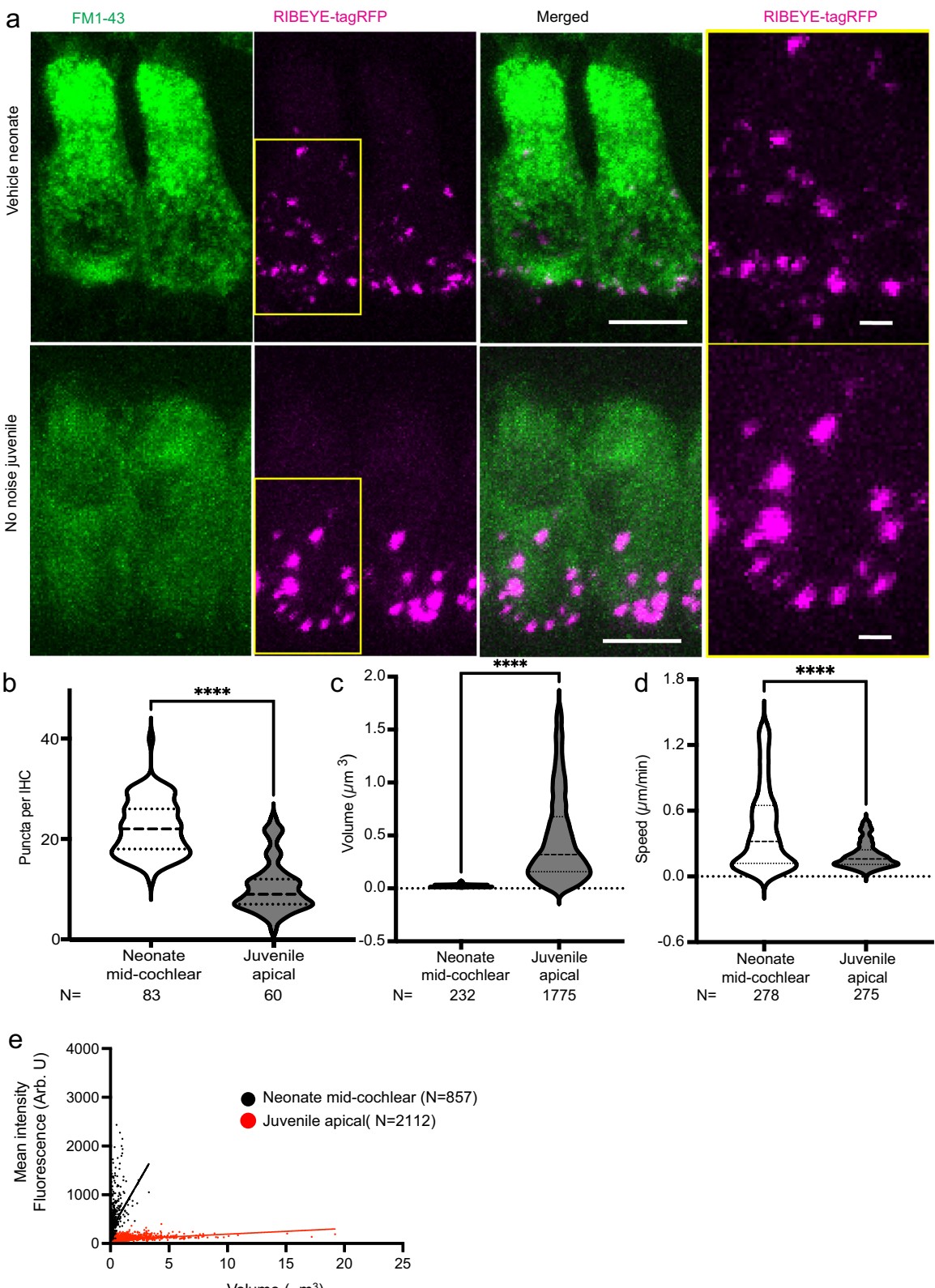

**Fig. 4 | Pre-synaptic ribbon characteristics in neonatal compared with juvenile mature-hearing cochlea. a** Representative live images of untreated neonatal (top) and juvenile, mature hearing (bottom) cochlea. Hair cells (FM1-43, green, left), RIBEYE-tagRFP (magenta, second from left), and merged images (third from left) are shown. Rightmost panel: magnified view of the basolateral region of individual hair cells with RIBEYE-tagRFP puncta from the yellow boxes in the second column. Scale bars = 5 μm (columns 1–3); 1 μm (rightmost column); **b–d** Comparison of number of puncta per IHC (**b**; $N$ = number of IHCs) and punctal volume (**c**; $N$ = number of puncta) and speed (**d**; $N$ = number of puncta) between neonatal (white) and juvenile (gray) cochlea. ****$p < 0.0001$. **e** Punctal volume was plotted against mean intensity for neonatal ($N$ = 857 puncta) and juvenile ($N$ = 2112 puncta) cochleae.

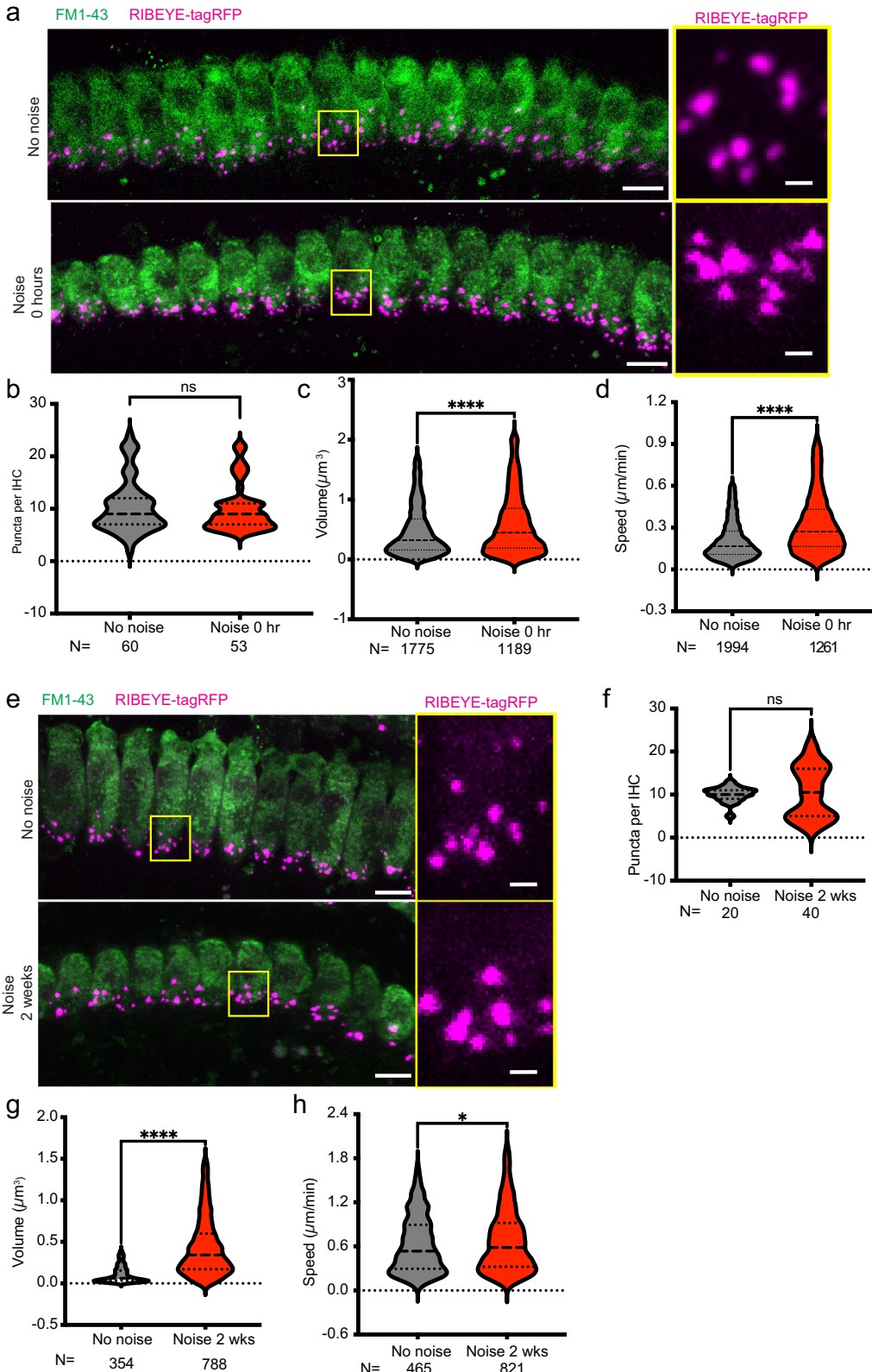

**Fig. 5 | Pre-synaptic ribbon changes after noise exposure. a** Representative live images of control (top) and noise-exposed (bottom) mature-hearing cochlear explants immediately after 2-h, 98 dB SPL noise exposure. Insets (right) of yellow boxes in low-magnification images show RIBEYE-tagRFP puncta (magenta). FM1-43 (green) marks the IHCs. Scale bars = 5 μm (left panels), 1 μm (right panels). **b** Number of RIBEYE-tagRFP puncta per IHC in control (gray) and noise-exposed (red) cochlear explants. *N* = number of IHCs. Comparison of volume (**c**) and speed (**d**) of RIBEYE-tagRFP puncta in control (gray) vs noise-exposed (red) cochlear explants. 3 cochleae were quantified for each condition, *N* = number of puncta. **e** Representative live images of control (top) and noise-exposed (bottom) mature hearing cochlea 2 weeks after noise exposure. Magenta: RIBEYE-tagRFP; Green: FM1-43 marking IHCs. Scale bars: 5 μm (left panels); 1 μm (right panels). Comparison of number of puncta per IHC (**f**; *N* = number of IHCs) and volume (**g**; *N* = number of puncta) and speed (**h**; *N* = number of puncta) of puncta in control (gray) and noise-exposed (red) cochlear explants. 3 cochleae (1 male, 2 female) for each condition were examined. *\**p* < 0.05; *\*\*\*\**p* < 0.0001; ns not significant.

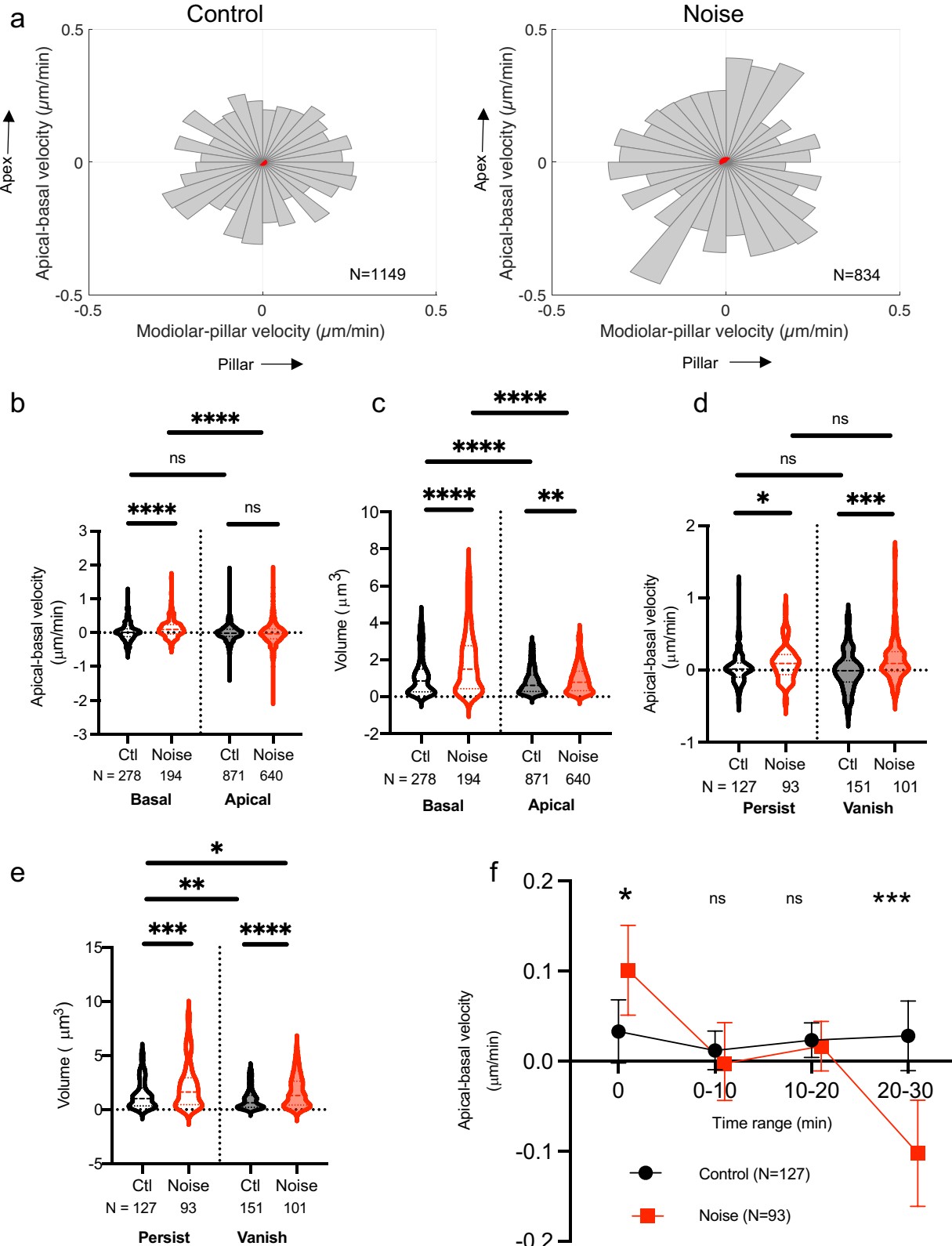

**Fig. 6 | Movement of pre-synaptic ribbons after noise exposure. a** Velocity vectors for pre-synaptic RIBEYE-tagRFP puncta were binned into 32 radial segments according to direction of movement (modiolar-pillar and apical-basal on the X and Y axes, respectively). For each segment, the mean speed is represented as the length of the segment. The 95% confidence interval for the mean velocity vector for all of the puncta analyzed for that condition is indicated by the red shaded area. **b, c** Basal-apical velocity and punctal volume for the 25th percentile basal-most and 75th percentile apical-most ribbons are represented as violin plots (dashed and dotted lines indicate median and IQR, respectively). **d, e** Among the 25th percentile basal-most puncta identified, noise induces movement towards the apex both in ribbons that subsequently vanish, where both persistent (persist) and transient (vanish) puncta exhibit increase in volume. **f** Among puncta that persist for 30 min, puncta from control, unexposed cochleae (black) show no overall movement, whereas those from noise-exposed cochlea (red) initially move towards the apex and then return towards to base. Points and error bars indicate means ± 95% CI. **b–f:** *$p < 0.05$; **$p < 0.01$; ***$p < 0.001$; ****$p < 0.0001$; ns not significant.

model for glutamate excitotoxicity in neonatal cochlear cultures as well as synaptopathic 98-dB SPL noise exposure in a juvenile, mature-hearing cochlear explant model.

At baseline, compared with neonatal cochlea, synaptic ribbons from mature hearing cochlea were less numerous, larger, and less mobile. The difference in ribbon volume, in particular, was large, with the mean (±SEM) volume of puncta in juvenile cochleae ($0.46 \pm 0.01 \ \mu m^3$) nearly 20-fold greater than in neonatal cochleae ($0.026 \pm 0.001 \ \mu m^3$, Fig. 4c). In a previous study using focused ion beam-scanning electron microscopy (FIB-SEM) to quantity ribbon volume during development from P9-P34, ribbon size was found to increase approximately 2-fold ($0.00344 \pm 0.00099$ versus $0.00568 \pm 0.00045 \ \mu m^3$)[40]. Measurements in P49 mice (more comparable in age to the juvenile mice used in our study) were larger still, ranging from $0.008–0.14 \ \mu m^3$ and $0.01–0.15 \ \mu m^3$ in two studies using serial block-face electron microscopy[48,49]. The absolute volumes measured in our study, using live fluorescent imaging, are not comparable to the much smaller structural measurements using electron microscopy; however, the trend towards larger ribbons in older cochleae is consistent. The much larger difference that we observed compared to previous studies may be related to differences in imaging and processing that were necessary between the neonatal and juvenile cochlear preparations, and is reflected in the significantly lower mean intensity of the punctal signals in juvenile animals (Fig. 4e).

After a 2-h exposure of neonatal cochleae to KA, which induces glutamate excitotoxicity, or exposure of mature hearing mice to 98 dB SPL, which induces cochlear synaptopathy, immediate physiological changes were observed. Neither chemical overstimulation nor noise exposure affected the number of pre-synaptic ribbons; however, noise exposure was associated with an immediate increase in synaptic ribbon volume in the mature hearing cochlea that was even more substantial 2 weeks later. The close correlation between the size and mean brightness of pre-synaptic puncta raises the possibility that observed changes in size may be related to changes in endogenous signal intensity, rather than actual volume increase; however, overall these findings on ribbon number and size corroborate previous observations from fixed specimens[37,38,40]. The ability from this live-imaging model to observe dynamic behavior of ribbon synapses revealed that both chemical overstimulation and noise exposure induces increased movement of pre-synaptic ribbons. Ribbons in KA-stimulated neonatal cochlea and noise-stimulated mature hearing cochlea both exhibited increased speed of ribbon movement. These initial increases in ribbon speed persisted, albeit to a lesser degree, 2 weeks after noise exposure. Ribbon speed in both neonatal and juvenile mouse cochlea was significantly slower than that observed in zebrafish using a similar transgenic system. In zebrafish, despite small spatial displacements over extended periods, short-term instability and high mobility of the ribbon's underlying structure was observed, challenging traditional views of ribbon physical rigidity and proposing a fluid mosaic model for the ribbon surface[33]. Further study in the mammalian system is required, especially improvement in temporal resolution, to further investigate whether these differences truly reflect species-specific differences in ribbon stability.

When the entire aggregate population of ribbons was examined together, there was no consistent directionality to the KA- or noise-induced ribbon displacement; overall, displacement vectors appeared to be randomly distributed. RIBEYE proteins are dynamic in nature but become stabilized in ribbon synapses[50]. The increased speed in the KA-treated and noise-exposed ribbons might be due to induction of instability in the ribbon synapses in the pre-synaptic zone. The lack of overall directionality suggests that ribbons are becoming unanchored; that is, RIBEYE anchorage is disrupted but without subsequent directed transport. In ribbon synapses, Bassoon anchors RIBEYE at the active zone of the presynaptic membrane. Although both Bassoon-deficient and wild-type IHCs have large RIBEYE spots at afferent synapses, Bassoon-deficient IHCs were also shown to possess unanchored and floating ribbons close to the synapses[51]. Furthermore, physical interaction between Bassoon and RIBEYE is thought to be present at the cytomatrix active zone in retina photoreceptors, maintaining the integrity of the ribbon complex[52].

Though, overall, ribbons did not have a concerted vector of displacement, a subgroup of ribbons in the mature hearing cochlea closest to the basal pole of the hair cells exhibited clear directionality. Compared to all ribbons in control, non-noise exposed animals as well as apical ribbons in noise-exposed animals, which exhibited no directionality of movement, basal ribbons moved towards the apex immediately after noise exposure. While it is possible that these basal ribbons in juvenile, noise-exposed mice are moving apically simply because they have no other direction to go after being unanchored by noise exposure, we did not observe this directionality in basal ribbons either in unexposed juvenile cochlea or in neonatal cochleae after KA exposure, suggesting that this directional movement is specific to noise-exposed mature hearing cochleae. This initial apical movement of the basal-most ribbons towards the hair-cell nucleus was followed by movement back towards the base of the hair cell, which may reflect re-attachment. These findings provide direct visual evidence of dynamic, directional movement of basal synaptic ribbons after noise exposure in the mature hearing cochlea.

Prior reports showed that synaptopathy occurs rapidly during noise exposure, with limited subsequent recovery[24], though strain-specific differences have been observed[53]. Synapses around the IHC are arranged based on fiber characteristics; large ribbons are associated with small receptor patches on the modiolar side, and small ribbons are linked to large receptor patches on the pillar side[54]. These gradients underlie variation in cochlear nerve response, reflecting a low-spontaneous discharge rate (SR) to high-SR gradient[55]. Following noise exposure, synapses are preferentially lost on the modiolar side of the IHC, and more orphan ribbons are found near the habenular end. These orphan ribbons appeared to recover eventually rather than degenerate. These conclusions, however, were all inferred from fixed imaging; it could not be determined or observed directly whether ribbons were migrating to and from the receptor patches, or whether they were being dissolved and reformed.

Our current study provides complementary evidence to directly show how ribbons are moving immediately after noise exposure. Though the overall population of ribbons appear to be moving in random directions, suggesting unanchoring, the basalmost subgroup of puncta migrated apically towards the nucleus and then returned to the cell membrane, suggesting that ribbons are recycled rather than immediately degraded and then reformed. Further refinement and study of this live imaging model may reveal further subgroups of ribbons that have characteristic behaviors after acoustic overstimulation that can be targeted for treatment of cochlear synaptopathy.

This study was limited by the spatial and temporal resolution of the imaging technique. Adequate 3D imaging of puncta required high-spatial-resolution laser-scanning confocal microscopy across the entire synaptic region, with each z-stack requiring 5 min to acquire on average, thus limiting temporal resolution. For large movements or rapidly drifting preparations, this might result in a systemic underestimation or overestimation of the punctal speed even after adjusting for the reference frame, if the imaged punctum was captured within a significantly different imaging plane from one frame to another. However, the preparations were quite stable, with minimal movement of the reference frame (Supplementary Fig. S1), and the punctal movements were small; therefore, it is likely that despite the prolonged acquisition time, the measured speeds are likely to be accurate at the 5-min time resolution limit. We are unable to make conclusions, however, about movement occurring faster than this limit; future studies restricting imaging to small volumetric regions of interest to enable faster image acquisition are needed to evaluate faster synaptic movement.

We relied on Imaris software to track puncta from one frame to the next; over a 20-min recording period with 6 serial images, less than half of puncta could be tracked across all images. These vanished puncta were not interpreted to represent actual loss of puncta over that time period, and the number of vanished puncta was not different between control and noise-exposed animals; however, it illustrates the challenges inherent to the imaging technique.

We observed other changes that occurred during routine culture and imaging: even in untreated cochleae the number of puncta detected in

neonatal cultures decreased after 24 h in culture, and the speed of punctal movement decreased over the first 20 min of serial imaging, (Fig. 3a, e). The reasons for these changes are unknown, but may reflect either a phototoxic response to the initial imaging itself or simply evolution of synapses in explant cultures that had previously been undetectable without serial imaging of the same preparation. These finding illustrate the necessity for consistency of culture duration and inclusion of rigorous control conditions in any study using neonatal cochlear cultures, whether using live or fixed imaging techniques.

While imaging of neonatal cochlear cultures enabled observation of ribbon dynamics during chemical overstimulation with KA, this is not a physiological perturbation; conversely, while we were able to examine ribbon dynamics after a physiologically relevant stimulus (noise exposure) in the mature hearing cochlea, we could not acoustically stimulate the explanted juvenile cochlea and directly observe the effect of sound stimulation, as was done previously in a limited fashion in the gerbil cochlear explant[56]. In future studies, further refinement of the imaging technique and mature hearing explant model can advance understanding of ribbon dynamics after noise exposure.

In summary, our findings suggest that noise exposure induces synapses to become more loosely anchored, and that a subpopulation of the basal-most synapses migrate apically, towards the nucleus. Within 20 min, some of these ribbons then migrate back towards the synaptic terminals, likely to re-form paired synapses. These insights suggest that further investigation of ribbon anchoring may uncover molecular targets for intervention to prevent this initial noise-induced ribbon dissociation or promote re-targeting in the early moments after noise exposure, thereby preventing permanent synaptopathy.

## Methods

### Cochlear explant dissection and culture
For neonatal cochlear cultures, postnatal day 4 (P4) to P6 C57BL/6 RIBEYE-tagRFP mouse cochleae (gift of Rachel O.L. Wong) of both sexes underwent dissection of the temporal bone in ice-cold Hanks' Balanced Salt Solution (HBSS). The membranous labyrinth was extracted from the modiolus, with removal of the stria vascularis and spiral ligament. Reissner's membrane and the tectorial membrane were removed. Explants were placed in the center of a 35 mm glass bottom cell culture dish coated with cell-tak and cultured overnight in Dulbecco's modified Eagle's medium (DMEM) with 10% fetal bovine serum (FBS) and 50 ug/mL ampicillin at 37°, 5% CO2.

For juvenile, mature-hearing cochlea, 7–10-week-old C57BL/6 RIBEYE-tagRFP mice of both sexes were euthanized with carbon dioxide and then decapitated. The temporal bone was extracted from the skull by removal of the auditory bulla and then placed in ice-cold Phosphate-buffered saline (PBS). Soft tissue and ossicles were removed with fine forceps. The bone covering the apical turn of the cochlea was flicked off with a #11 scalpel, preserving the membranous labyrinth and exposing the helicotrema. Reissner's membrane was removed (Supplementary Fig. S5)

### Immunohistochemistry
P4-P6 RIBEYE-tagRFP neonatal mouse cochleae were cultured overnight and fixed the next day with 4% paraformaldehyde at room temperature (RT) for 20 min. After 3 PBS washes, and blocking (94% PBS, 5% normal goat serum, and 1% Triton X-100, 30 min at RT), cultures were incubated in primary (24 h, RT) and secondary antibodies (1.5 h, RT) in 98.6% PBS, 1% normal horse serum, and 0.4% Triton X-100. Primary antibodies were as follows: 1) mouse (IgG1) anti-CtBP2 (1:200; #612044, BD Transduction Labs) for pre-synaptic ribbons; 2) mouse (IgG2a) anti-GluA2 (1:2000; #MAB397, Millipore) for post-synaptic receptor patches; 3) rabbit anti-Myosin 7a (1:400; #25-6790 Proteus Biosciences) for hair cells; and 4) chicken anti-NF-H polyclonal (1:1000; Chemicon). Secondary antibodies: 1) Alexa Fluor 488-conjugated goat anti-mouse (IgG2a; 1:1000; #A21131, Life Technologies); 2) Alexa Fluor 568-conjugated goat anti-mouse (IgG1; 1:1000; #A21124, Life Technologies); 3) Alexa Fluor 647-conjugated chicken anti-rabbit; (1:200; #A21443, Life Technologies); and 4) Alexafluor 488-conjugated goat anti-chicken (1:1000). Confocal (Nikon A1R upright line-scanning) microscopic images of the hair-cell region in the mid-portion of the middle cochlear turn were obtained with 60X oil immersion objective (Plan Apo, 1.40 NA).

### Live imaging
Neonatal cochlear cultures were incubated for 24 h and then loaded with FM1-43 dye (5 μM for 15 s) with three 10-sec HBSS washes and incubated for 30 min at 37 degrees for de-esterification to delineate hair cells.

Juvenile cochlear explants were dissected as described above and then immediately loaded with FM1-43 dye (7.5 μM for 10 s) with three 10-s HBSS washes and incubated for 5 min at 37 degrees to delineate hair cells. Specimens were placed on a custom 3D printed slide and suspended in HBSS in a hole made on a plastic coverslip for live imaging. Time from euthanasia to initiation of imaging averaged 15 min.

Volumetric regions of interest of the organ of Corti with one intact row of IHCs and three intact rows of outer hair cells (OHCs) were visualized. Optical sections in the x–y plane were recorded at 2x averaging, 1.2 AU pinhole, 1.1 μs/pixel dwell time, 0.257 μs frame rate, and 0.4 μm intervals in the z-axis spanning the entire IHC height in the middle cochlear turn (neonates) and apical cochlear turn (juveniles) on a Nikon A1R upright line-scanning confocal microscope with 60x water-immersion objective (NIR Apo, 1.0 NA) and stagetop incubator (OKOlab). All solutions were pre-warmed to 37 degrees, with incubation and imaging controlled for temperature. CO₂, humidity, pressure, and air flow were controlled for neonatal cultures only. Acquisition of the entire confocal Z-stack took approximately 5 min. For dynamic measures of displacement and speed, a pair of Z-stacks was performed one immediately after the other, thus resulting in an average minimum time interval between images of 5 min, which determined the temporal resolution of live timecourse imaging. For experiments in which dynamic measurements at multiple timepoints 2–24 h apart, cochleae were returned to the CO₂ tissue-culture incubator in between imaging sessions. For continuous serial measurements, image stacks were acquired continuously with the specimen maintained in the stagetop incubator.

To compare puncta imaged in live and fixed specimens, neonatal cochleae were cultured overnight and two live successive z-stack images were captured. Cochleae were immediately fixed with 4% PFA and hair cells stained with rabbit anti-Myosin 7a (1:400; #25-6790 Proteus Biosciences) and Alexa Fluor 647-conjugated 126 chicken anti-rabbit; (1:200; #A21443, Life Technologies). These fixed and stained cochleae were then imaged using identical parameters as for live imaging.

### Image processing
RIBEYE puncta were rendered using the Spots function on Imaris (v9.9.1). A volumetric region of interest containing the IHCs was manually segmented. Estimated X-Y Diameter (0.400 μm), Quality (116 for neonates and 85 for adults) and Region Threshold (300 for neonates and 85 for adults) were determined initially from manual validation; these parameters yielded spot identification that approximated puncta identified manually in pilot imaging experiments. Parameters were determined separately for neonatal and juvenile experiments, and were kept constant across all age-matched experiments. Auto depth placement was based on the FM1-43-labeled IHCs in the FITC channel. The Surface function was used to create 3D-rendered hair cells. The Smooth option was selected with the Surface detail set to 0.4 μm, followed by appropriate thresholding using the Absolute intensity parameter. Next, the filter "Number of voxels img=1" was used to select all the correct hair cell surfaces. This enabled the establishment of a 3D reference frame with constant IHC placement, upon which movement of the RIBEYE puncta could be ascertained. After punctal identification and tracking, we extracted, for each timepoint (for static readouts) and successive pairs of timepoints (for dynamic readouts): punctal volume and X, Y, and Z position, from which we calculated displacement over time (velocity). Workflow for image processing is shown in Supplementary Fig. S6. To quantify the number of RIBEYE puncta per IHC, one blinded investigator performed manual count of the number of RIBEYE puncta per IHC[44].

### Noise exposure and glutamate excitotoxicity

As a model for glutamate excitotoxicity, neonatal cochlear explants were dissected, incubated for 24 h, and exposed to culture media containing 0.4 mM KA for 2 h. Specimens were subjected to live imaging before and during treatment as well as 2 and 24 h post-treatment.

As a model for cochlear synaptopathy, 7–10-week-old RIBEYE-tagRFP mice were exposed to 8–16 kHz octave-band white noise at 98 dB SPL for 2 h[44]. Noise exposure was performed on awake mice in a custom-built, calibrated, reverberant sound chamber. Animals were placed freely in an acoustically transparent wire cage on a rotating platform for the duration of the noise exposure. Unexposed littermates were used as negative controls. Auditory brainstem response (ABR) thresholds were measured in open-field response to pure-tone pips at 8, 16, and 32 kHz (RZ6, Tucker-David Technologies) in a soundproof chamber[44].

### Statistical analysis and quantification

Data analysis was performed using Matlab (R2023a, Mathworks) and Prism (v9, GraphPad). Pairwise comparisons were conducted using Student's *t* test (two-tailed, paired or unpaired when appropriate) with data presented as means ± 95% CI. Alpha value of 0.05 was used to determine statistical significance. Outlying values were removed when indicated as determined by the ROUT method (Prism) with Q value set to 1%.

### Study approval

Experimental protocols were approved by the University of California, San Francisco Institutional Animal Care and Use Committee, and all methods were carried out in accordance with relevant guidelines and regulations. We have complied with all relevant ethical regulations for animal use.

### Reporting summary

Further information on research design is available in the Nature Portfolio Reporting Summary linked to this article.

### Data availability

All data supporting the findings of this study are available within the paper and its Supplementary Information. The source data behind the graphs in the paper can be found in the Supplementary Data file.

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

## Author contributions

Initial design: N.I.M., P.S., D.K.C. Experimental and ethical oversight, and funding: D.K.C. Experimental contributions: N.I.M., P.S., Y.P., I.R.M., E.T., D.K.C. Data and statistical analysis: N.I.M., P.S., Y.P., D.K.C. Manuscript drafting: N.I.M., P.S., D.K.C. Manuscript review and approval: N.I.M., P.S., Y.P., I.R.M., E.T., D.K.C. Multiple first and corresponding authors: N.I.M. and P.S. share first position. N.I.M. was primarily responsible for experimental design, development of the experimental preparation and design and conduct of the imaging experiments. P.S. was primarily responsible for development of the analytic process and conducted the majority of the data analysis. Between N.I.M. and P.S., N.I.M. is listed first due to being the original designer of the project.

## Competing interests

The authors declare no competing interests.
