## [Peer Review File · Communications Biology]

Reviewers' comments:

Reviewer #1 (Remarks to the Author):

This is a very interesting study where the authors performed live imaging in cochlear explants of RIBEYE-tagRFP mice and examined the dynamic change of synaptic ribbons in the inner hair cells, before and after insults known to cause hidden hearing loss. The approach is straight-forward and novel, and the raw data are of high quality. The presentation is good but not great, as there are places where changes can be done readily to add value to the whole package. In addition, clarifications can be added to the main text to improve readability of the manuscript.

Major points:

1, In the introduction, to set up a more complete context, I suggest to add a line of studies using fluorescence-tagged CTBP2-binding short peptides for live imaging of synaptic ribbons (delivered through patch-clamp pipettes, pioneered by David Zenisek in retina, but adopted in a number of studies on hair cells), and also a 2017 study by Katie Kindt from NIDCD (RIBEYE-tagEGFP, on zebrafish neuromast hair cells). In addition, please discuss the latter in the discussion, as ribbons in the zebrafish neuromast hair cells seem to be 10 times more mobile.

2, Figure 1B, it caught my attention that the presynaptic and postsynaptic puncta seem to be quite far apart for each pair, please explain. I am under the impression they are supposed to be opposed to each other quite closely.

3, Line 242-245, the two parts of this sentence has a gap, which is supposedly to be something like “but there was no significant difference in the number of ribbon puncta in 24h after vehicle and KA treatment, which is consistent with ...”. Also, the authors need to address the decline in the number of ribbon puncta before and after vehicle treatment. What are the possible causes? Also, for Figure 3A, 3B and 3C, please change or remove statistic comparisons in B and C so it aligns better with A (they are from the same raw data, the logic shall be the same).

4, Figure 3C, I would like to see the same measurement in fixed tissue. The movement is very subtle, so small vibration or drifting could contribute substantially. Additionally, repetitive imaging could cause bleaching, so the shape of ribbons can be slightly different, which can be mistakenly taken as movement, depending on how the position was pinpointed from the fluorescence signal.

5, Figure 3E, why is there a decline in velocity for both groups while the acquisition is in progress? Please explain. In addition, given that the velocity is changing along the time, at what time did the authors use it for Figure 3C? Or is it the average of all times? Please specify it either in the methods or the results.

6, Figure 4C, I guess I am surprised that the volume of ribbon puncta increased almost 10 times from P6 to 7 weeks. The authors need to discuss this in the discussion, compare and contrast their results with those in published studies, in a quantitative manner.

Minor points:

1, Figure 2C, I would suggest to put the raw data on the top, and the 3D rendering on the bottom, and also add moving paths of 2 or 3 representative ribbons (simple dots and lines are sufficient). Currently, the ribbons all look very stationary in the figure.

2, Line 238-240, to avoid confusion with Line 243-245, I suggest to expand this sentence, to something like “the number of ribbon puncta remained unchanged from the control (my guess, please check), but the number of postsynaptic puncta was reduced, making the number of paired synapses reduced following KA treatment, consistent with previous studies.”

3, Figure 3A, 3B and 3C, please remove “24h” from “24h pre”, unless live imaging was indeed performed 24h before the drug treatment (if this is the case, please say so explicitly in the methods). My understanding is that the control data was obtained not long before the drug treatment, if not immediately before. Also, I would like to suggest to change “Control” to “Vehicle”, and say explicitly what is “Vehicle” (de-ion water, saline, DMEM?) in the methods.

4, Line 275-276, to align better with the other section titles, I suggest to change this title to something like “RIBEYE-tagRFP puncta are more mobile in noise-exposed juvenile, mature-hearing cochlear cultures”.

5, Line 306-308, I suggest to delete this sentence, because you are not making a claim about high vs. low spontaneous-rate fibers. If you do, I suggest to choose ribbons in the quarter on the base and toward the outer hair cells.

6, Line 443, I would like to see the discussion of how an acquisition time of 5 mins could affect the measured velocity of moving ribbons, does it cause underestimation or overestimation?

Reviewer #2 (Remarks to the Author):

The study of Mohamad and colleagues addresses an important question of synaptic ribbon dynamics after noise exposure. The authors address this question by using a novel tool, cochlear explant expressing fluorescently-tagged ribbons, of mice subjected to noise exposure just prior to imaging. The authors first validate the model using neonatal cultures exposed to KA to evoke direct glutamate excitotoxicity and then expand the model to the mature organs from hearing animals. The authors show that both, direct glutamate excitotoxicity as well as noise overexposure induce increase in ribbon volume, unanchoring from synaptic tethers and increased mobility of the ribbons. Upon noise exposure some ribbons seem to move towards nucleus and back, however there seems to be no overall reduction in the number of the ribbons, neither upon KA treatment nor upon noise exposure. I would like to congratulate the authors on performing these technically challenging experiments. This is a very

important development of the technique that should allow to address interesting and important questions on the mechanisms of noise noise-evoked synaptopathy in the future. The current limitation of the study and technique is how long the acute preparations can be maintained and imaged and a very poor time resolution. Nevertheless, the MS opens a path for further improvements of the methodology and is thus very important and novel in my view. Furthermore, even with the limited time resolution, it does reveal some interesting consequences of glutamate excitotoxicity and noise overexposure on the presynaptic ribbons. I have a few comments on the present work.

Major:

1. As mentioned above, the main limitation of the study is the limited time resolution. Nevertheless, if I am not mistaken, the fact that the ribbon punctae were tracked only once every 5 minutes is only revealed at the very end of the discussion. Since this knowledge is fairly important for judging the parameter estimations such as speed and velocity, it would be fair to readers to state that clearly somewhere at the beginning of the MS and also to be a bit more conservative in how reliable and valid the assessment of the velocity/speed of puncta is. If the majority of the spots seem to “move” randomly, the “speed” of puncta movements may be significantly underestimated with an imaging limited to every 5 minutes. I understand that this was the current technical limitation of the study as the cells were imaged throughout the whole volume and the method was just recently established for these purposes, still, it would be fair to state this much earlier and add appropriate cautionary statements.
2. The loss of ribbons within 24 hours of vehicle treatment seems quite substantial, judged from the images perhaps 25-30%. The explants were cultured for 24h prior to the first imaging, if I understand correctly. I assume the cultures kept losing the ribbons for at least the next 24 hours, is this degeneration due to culturing? Please, comment in the MS. Is it so that mostly the very small ribbons disappear (when looking at the violin plot in B, the small “balloons” below the large center one seem to disappear or thin out?)? Where are these cultures kept between the imaging sessions? Please, specify in the methods.
3. What does immediately after KA treatment mean? Immediately after application of KA or immediately after stop of KA treatment (t.i. immediately after 2h KA application)? Based on the speed on Figure 3E, I assume this must mean immediately after 2h KA-application (t.i. immediately after “KA removal”), or? Please, specify. Are those 5 representative timecourses per condition (I assume the authors recorded many more?)?
4. From the panels F-G in the Figure 3 it seems like the KA-group does not move more neither in the apical-basal nor in the modiolar-pillar direction. If anything, it is the other way round. So, it seems like the KA-treated ribbons move more, but less in the one or the other given direction, as also the authors point out. But, does that mean that they move more randomly or just along some other trajectories, which are along neither of the two investigated axes? Have the authors looked at the mean square displacement of the puncta over time to identify how much the ribbons travel and in what way?
5. Again, I would like to congratulate the authors on tracking the ribbons in the mature cochleae. From the quality of the prep to the control of movement and correction for that, the task must have been anything but trivial. It would thus be very nice if the authors explained a bit more in the methods on the technical aspects of the study of how they corrected for the potential movements of the preps. In the acute explants, I can very well imagine that the cells may not be static and possibly also change the shape in the course of the experiment. If that is the case, how was the image/individual cell registered in that case and what was taken as the reference point(s) based on which the ribbon movement was

calculated? The authors speak about a stable 3D framework. How stable is that? With other words, while FM1-43 is a great reference, how did the authors account (correct) for the coordinates if the cellular shape changed? Or did that not happen? Also, connected to that, were the organs somehow fixed or did the slide and the coverslip with the customized hole provide good “fixation” of the preps in the chamber? I am not sure if that is explained well enough. And did that actually prevent the issues above, like shape change, large movements etc...? It would be nice if the authors gave some more information on that.

6. The animals were exposed to 8-16kHz noise band at 98 dB SPL for 2 hours, which is a noise paradigm often used in the field. Since there is variability between the results obtained in different labs and different conditions (based on background, age, experimental setups, gender etc), did the authors check the consequences of this exposure on hearing in their animals in these or prior experiments (ref)? Is it causing the TTS in their animals? Or, did the immunohistochemistry reveal a typical loss of synapses in the midcochlear and basal regions?

7. Interestingly, 2h 8-16kHz 98 dB exposure in 8w-old CBA male mice caused a temporary loss of ribbons in the apical cochleae (Liberman et al., 2015), which recovered within 24h. The data of the present study does not seem to show that. It would be interesting to see how well these studies compare (e.g. in terms of hearing damage or loss of synapses in a more basal part of the cochlea, see my previous point).

8. What I furthermore find interesting, is, that while the average number of puncta per IHC do not change 2w after exposure, the variance seems to increase a lot. Have authors thought of a potential meaning to that, e.g. loss vs repair and (over-)repair?

9. Please, specify the gender of the animals used for noise exposure.

10. I do not intend to be overly critical, just an observation: The authors suggest that their “basal” puncta based on the orientation should correspond to the ribbons connecting the HSR fibers and the “apical” then, I assume, the LSR and MSR fibers (?). If so, I would expect the basal (HSR) puncta to be smaller rather than larger than the other. Or, to be precise, if anything, not to see a difference as that is in mice anyway not so large. However, the authors seem to see the opposite. How do you comment on that, a coincidence?

11. Are the puncta that vanish randomly distributed or do they belong to one or the other group (modiolar-pillar etc.)?

12. Figure 6F: I do not understand the x-axis of the figure, e.g. how is the velocity at point 0 determined? If I understand correctly, the positions of the puncta were determined every 5 minutes, so the authors actually look at how much the puncta moved from 0 to 300 s, from 300 to 600s etc.. Is that what this image shows? If not, would that not be the most straightforward to show? Also, would it not make more sense to rather display the mean displacement of the puncta rather than velocity?

Minor:

1. Methods, line 139, and results line 264. “Time from euthanasia to initiation of imaging averaged 10 minutes.” (results: “...within 10 minutes”) If all given times are correct, is that realistic? I suppose we can assume that loading with dye and three HBSS washes have taken a minute, 5 minutes was reportedly incubation in 37°C, which adds up to 6 minutes. So, the experimenter was left with 4 minutes in total to prep carefully, mount the explant on the slide and under the microscope, and find the perfect position to image under the microscope and initiate a recording. And that on average? Or even maximally (as stated in the results)?

2. Methods, line 143: "1.1 dwell time". Time unit?
3. Line 245: Do the authors want to say not affected by glutamate excitotoxicity within 24 hours? Significant losses of ribbons were observed previously after longer times, e.g. a few days after KA exposure, right?
4. Line 243: correct to 24h after KA "or vehicle" treatment, or something similar
5. Figure 3: I assume all "24h pre" labels should be changed to "pre". Or were they performed 24h before treatment?
6. Figure 4: it might perhaps be wise to add midcochlear and apical to the labels of the panels B-D as that in addition to the age difference may be a contributor to the difference in the number of puncta?

Reviewer #3 (Remarks to the Author):

Mohamad and colleagues present a study on the influence of medium noise trauma on inner ear ribbon synapses, as observed using a fluorescent marker. They find that synaptic ribbons appear to become more mobile in organotypic cultures of immature organs of Corti treated with kainic acid as well as in acute explants from noise-treated hearing mice. In the latter, they also find increased size of ribbons after noise trauma, as well as an apparent directional movement of ribbons from the basal pole of the inner hair cells. The increased size and mobility of ribbons persist even two weeks after noise stimulation.

The findings are interesting and well-presented, with sufficient detail to recreate the experiments and adequate analysis, but they do not present a major gain of new information. It has been described before that ribbons appear to move away from the synapse following noise exposure and that ribbon size increases (e.g. Kujawa and Liberman, *J Neurosci* 2009; Liberman et al., *JARO* 2015). As the authors note, this is however the first time that this has been described in living, non-fixed cells. As such, it is a confirmation of existing data using other means, which is also an important contribution, so I support publication.

There are however some parts of the manuscript which are either not clear to me or could be improved:

- I think the conclusions about directional movement of ribbons should be discussed more carefully. The observation is that ribbons from the basal pole of the hair cell have a higher tendency to move towards the apex after noise trauma. However, since the bottom end of the hair cell is restricting downwards or sideways movement (simply because it is a cul-de-sac) detached ribbons are probably more likely to move in an apical direction, simply because this is the only way to go. In this regard the data in Fig. 6F might also indicate that ribbons re-attach after ~20 minutes, so that after moving down they stop moving, which would result in on average increased downward movement.

- Following up on the theory that the higher undirected speed of ribbons originates from detached ribbons, it would be interesting to relate the movement speed with the distance of the identified puncta from the hair cell membrane (which was nicely reconstructed, as visible in Fig. 2B).

- It would be interesting to assess the relationship between size of the ribbons and the integrated fluorescence intensity from the ribbon volume. This would indicate if larger ribbons meant additional RIBEYE material or just less dense (“puffy”) ribbons. The authors remark about a “close correlation between the size and mean brightness of pre-synaptic puncta” (line 379), I am not quite sure what they are referring to here, since I did not find that data in the manuscript.

- Why was no imaging performed during the application of KA?

- It is stated that KA treatment decreased the number of paired synapses. How were paired synapses identified? Figure S2 does not make it obvious how one could identify them with the large amount of GluR2 spots visible.

- The authors validate their technique. It would help here if they would also show examples of which puncta were recognized as spots by their Imaris settings and which weren’t. Figs. 2A-C give a rough indication, but are not clear.

Minor points:

- It is not quite clear to me whether the experiments on juvenile mice were performed in semi-intact cochleae or organ of Corti explants. I assume the former, but this should be stated more clearly.

- Line 86: The authors state that ribbons are not required for establishing synapses in vivo in the retina. The same is true for inner hair cells, as shown by Becker et al. and Jean et al., both eLife 2018. These should be cited.

- Line 100: “gift of R.L.” should be spelled out.

- Line 117/118: What do the abbreviations PBT1 and PBT2 mean?

- Line 143 “1.1 dwell time” lacks the unit (μs ?). Also, it would be helpful to indicate the frame rate (assuming that it was identical in all experiments).

- Line 158 “Auto depth placement was based the FM1-43...” should be “based on the FM1-43”.

- Line 210: 75% colocalization, where did this number come from? Is this based on a threshold for pixel intensities?

- Line 268: “significantly higher (Figure 4C), consistent with prior reports” should better be something like “significantly higher (Figure 4C). Both observations are consistent with prior reports”

- Line 437-439: If the stacks took at least 5 minutes to acquire, how can there be 6 images in a 20 minute recording?

- Line 480-481: "top2 and "bottom" should be "left" and "right"

- Figure 1F: The axes should have units, even if it is only "fluorescence intensity (a.u.)"

- Figure 2B, C: What are the grey boxes in the bottom of the images?

- Figure 6F: Why are exact times presented for the first and last point, but time ranges for the two in the middle? Movement can only occur between time points, so points 1 and 4 should also indicate a range?

Thank you very much for the feedback. We have thoroughly revised the manuscript and provide point-by-point responses below in blue.

Reviewers' comments:

Reviewer #1 (Remarks to the Author):

This is a very interesting study where the authors performed live imaging in cochlear explants of RIBEYE-tagRFP mice and examined the dynamic change of synaptic ribbons in the inner hair cells, before and after insults known to cause hidden hearing loss. The approach is straight-forward and novel, and the raw data are of high quality. The presentation is good but not great, as there are places where changes can be done readily to add value to the whole package. In addition, clarifications can be added to the main text to improve readability of the manuscript.

Major points:

1, In the introduction, to set up a more complete context, I suggest to add a line of studies using fluorescence-tagged CTBP2-binding short peptides for live imaging of synaptic ribbons (delivered through patch-clamp pipettes, pioneered by David Zenisek in retina, but adopted in a number of studies on hair cells), and also a 2017 study by Katie Kindt from NIDCD (RIBEYE-tagEGFP, on zebrafish neuromast hair cells). In addition, please discuss the latter in the discussion, as ribbons in the zebrafish neuromast hair cells seem to be 10 times more mobile.

Thank you for pointing out these important studies. We have addressed this as requested, in the Introduction:

“Live imaging of perturbed synapses helps fill in these gaps in knowledge. Work in goldfish retina bipolar cells in which a short peptide with affinity for the CtBP subunit of the RIBEYE protein – the major component of ribbons — or, later, calcium indicators, were introduced via whole-cell patch pipette, provided critical insight into the development and organization of synaptic ribbons^{27,28}. Subsequent development of a mouse transgenic line in which RIBEYE is tagged by a red fluorescent protein (RIBEYE-tagRFP), enabled visualization of the dynamic assembly of ribbon synapses during synaptogenesis and maturation in the mouse retina²⁹. This study showed that the endogenous, genetically encoded RIBEYE-tagRFP signal is an effective tool for tracking the development of synaptic ribbons, and that while synaptic ribbons can affect the stability of nascent bipolar cell synapses, they are not essential for establishing these synapses in vivo.

Similarly, in RIBEYE knockout mice lacking synaptic ribbons in hair cells, only minor disruptions in the function of hair-cell synapses and minor auditory impairments are present, suggesting the existence of compensatory mechanisms in this model system³⁰⁻³². However, RIBEYE was shown to play a crucial role in organizing presynaptic CaV1.3 calcium channels, although the localization of these channels at the hair-cell synapse remained unaffected. In the absence of RIBEYE, numerous small CaV1.3 clusters were observed at each synapse, deviating from the usual single organized structure^{30,32}.

Initial imaging studies of live ribbon synapse activity predominantly relied on dissociated retinal cells to explore structural ribbon dynamics. RIBEYE-tagEGFP transgenic zebrafish have been used to investigate hair cell ribbons in the lateral line system over extended periods without compromising the structural integrity, osmolarity, or mechanical forces surrounding the cells – factors that could influence calcium influx and synaptic activity³³. Discrepancies with previous studies highlight the potential for advanced imaging technologies to provide deeper insights into the dynamic nature of ribbon synapses and vesicle interactions in different cell types and experimental conditions.”

And in the Discussion:

“Ribbon speed in both neonatal and juvenile mouse cochlea was significantly slower than that observed in zebrafish using a similar transgenic system. In zebrafish, despite small spatial displacements over extended periods, short-term instability and high mobility of the ribbon’s underlying structure was observed, challenging traditional views of ribbon physical rigidity and proposing a “fluid mosaic” model for the ribbon surface³³. Further study in the mammalian system is required, especially improvement in temporal resolution, to further investigate whether these differences truly reflect species-specific differences in ribbon stability.”

2, Figure 1B, it caught my attention that the presynaptic and postsynaptic puncta seem to be quite far apart for each pair, please explain. I am under the impression they are supposed to be opposed to each other quite closely.

Because we are showing maximum-intensity projections, there are a number of pre-synaptic puncta and post-synaptic densities represented that are not paired, obscuring the paired ones. When actually counting paired synapses, we look at the confocal slices and 3D volumes. We have added an inset to Figure 1B to illustrate an example of a paired synapse from a single confocal slice.

3, Line 242-245, the two parts of this sentence has a gap, which is supposedly to be something like “but there was no significant difference in the number of ribbon puncta in 24h after vehicle and KA treatment, which is consistent with ...”. Also, the authors need to address the decline in the number of ribbon puncta before and after vehicle treatment. What are the possible causes? Also, for Figure 3A, 3B and 3C, please change or remove statistic comparisons in B and C so it aligns better with A (they are from the same raw data, the logic shall be the same).

Thank you for this comment; we realize the text was confusing as written. We intended to convey that both control and KA-treated cochleae had decreases in ribbon number after 24h in culture, and that these decreases were not statistically significantly different. We clarified this in the Results as follows:

“Control cochleae exhibited a decrease in the number of RIBEYE-tagRFP puncta after 24 hr in culture. Cochleae treated with KA also showed a decrease in puncta number; the number of puncta, however, was not statistically significantly different between control and KA-treated cochleae at any timepoint. (Figure 3A). This is consistent with prior findings that the number of pre-synaptic ribbons is not affected by glutamate excitotoxicity within 24 hours^{25,41}.”

We then added a section to the Discussion: Limitations addressing this baseline decrease in synaptic ribbon number in neonatal cultures:

“We observed other changes that occurred during routine culture and imaging: even in untreated cochleae the number of puncta detected in neonatal cultures decreased after 24h in culture, and the speed of punctal movement decreased over the first 20 minutes of serial imaging, (Figure 3A,E). The reasons for these changes are unknown, but may reflect either a phototoxic response to the initial imaging itself or simply evolution of synapses in explant cultures that had previously been undetectable without serial imaging of the same preparation. These finding illustrate the necessity for tight control of culture period and inclusion of appropriate control conditions in any study using neonatal cochlear cultures, whether using live or fixed imaging techniques.”

Finally, we aligned the statistical comparisons in Figure 3B/C with Figure 3A, as recommended, and adjusted the manuscript text accordingly.

4, Figure 3C, I would like to see the same measurement in fixed tissue. The movement is very subtle, so small vibration or drifting could contribute substantially. Additionally, repetitive imaging could cause bleaching, so the shape of ribbons can be slightly different, which can be mistakenly taken as movement, depending on how the position was pinpointed from the fluorescence signal.

Thank you for this comment. To address this, we performed speed measurements in a live cochlea, fixed it, and then performed the same measurement in the fixed specimen. We repeated this three times; each time, the measured speed was indeed higher in the live specimen, confirming that the movement we are seeing is unlikely to be entirely due to optical or mechanical artifact. We added this data to a new **Supplemental Figure S5**, accompanied by this manuscript text:

“We then assessed movement of synaptic ribbons by measuring the displacement of puncta from one frame to the next and dividing by the time interval. Comparison of speed measurements in live-imaged cochleae and the same cochleae after fixation confirmed that the live-imaged puncta had greater movement, thus excluding imaging or mechanical artifact (Figure S5).”

5, Figure 3E, why is there a decline in velocity for both groups while the acquisition is in progress? Please explain. In addition, given that the velocity is changing along the time, at what time did the authors use it for Figure 3C? Or is it the average of all times? Please specify it either in the methods or the results.

The decline in velocity for both groups over the first 20 minutes of imaging may be a consequence of the imaging itself or incubation conditions. While we do control temperature, CO₂, and humidity using a stagetop incubator, the imaging itself may induce some phototoxicity or temperature changes in the preparation that we can completely account for. We address this in the Discussion: Limitations together with our discussion of decline in ribbon number after 24h of imaging; these changes that we were seeing using live imaging may be artifacts of the culture and imaging process itself, and highlight the necessity for appropriate controls:

“We observed other changes that occurred during routine culture and imaging: even in untreated cochleae the number of puncta detected in neonatal cultures decreased after 24h in culture, and the speed of punctal movement decreased over the first 20 minutes of serial imaging, (Figure 3A,E). The reasons for these changes are unknown, but may reflect either a phototoxic response to the initial imaging itself or simply evolution of synapses in explant cultures that had previously been undetectable without serial imaging of the same preparation. These findings illustrate the necessity for consistency of culture duration and inclusion of rigorous control conditions in any study using neonatal cochlear cultures, whether using live or fixed imaging techniques.”

We also clarified that the speed measurements in Figure 3C (before and 24h after KA treatment) were obtained from single pairs of images, whereas the measurements shown in Figures 3D and 3E are from serial images taken during KA treatment. Figure 3D shows the full distribution of speeds immediately after KA treatment (t=0), and Figure 3E shows the full timecourse. We clarified this as follows:

“These speed measurements were performed using a single pair of images before and after KA treatment. We then performed serial timecourse imaging during KA exposure. The increase in speed was present immediately after KA exposure (Figure 3D) and remained elevated over a 1-hr recording period (Figure 3E).”

6, Figure 4C, I guess I am surprised that the volume of ribbon puncta increased almost 10 times from P6 to 7 weeks. The authors need to discuss this in the discussion, compare and contrast their results with those in published studies, in a quantitative manner.

Thank you for this comment. We too were surprised by the large difference. While previous studies with much more definitive volume measurements (using scanning electron microscopy) do show an increase in ribbon size with age, the changes are much smaller. It is likely that the magnitude of difference that we are seeing is additionally due to differences in imaging conditions between the neonatal and juvenile cochlear preparations. We have added the following paragraph to bring in some quantitative comparisons to previous studies and acknowledge the likely contribution of these imaging differences:

*“At baseline, compared with neonatal cochlea, synaptic ribbons from mature hearing cochlea were less numerous, larger, and less mobile. The difference in ribbon volume, in particular, was large, with the mean (\pm SEM) volume of puncta in juvenile cochleae ($0.46 \pm 0.01 \mu\text{m}^3$) nearly 20-fold greater than in neonatal cochleae ($0.026 \pm 0.001 \mu\text{m}^3$; **Figure 4C**). In a previous study using focused ion beam-scanning electron microscopy (FIB-SEM) to quantify ribbon volume during development from P9-P34, ribbon size was found to increase approximately 2-fold (0.00344 ± 0.00099 versus $0.00568 \pm 0.00045 \mu\text{m}^3$)⁴⁸. Measurements in P49 mice (more comparable in age to the juvenile mice used in our study) were larger still, ranging from 0.008 - $0.14 \mu\text{m}^3$ and 0.01 - $0.15 \mu\text{m}^3$ in two studies using serial block-face electron microscopy^{49,50}. The absolute volumes measured in our study, using live fluorescent imaging, are not comparable to the much smaller structural measurements using electron microscopy; however, the trend towards larger ribbons in older cochleae is consistent. The much larger difference that we observed compared to previous studies may be related to differences in imaging and processing that were necessary between the neonatal and juvenile cochlear preparations, and is reflected in the significantly lower mean intensity of the punctal signals in juvenile animals (**Figure 4E**).”*

Minor points:

1, Figure 2C, I would suggest to put the raw data on the top, and the 3D rendering on the bottom, and also add moving paths of 2 or 3 representative ribbons (simple dots and lines are sufficient). Currently, the ribbons all look very stationary in the figure.

We switched the positions of the raw/rendered data for Figure 2D (formerly 2C; we added a middle panel 2C illustrating which puncta are recognized and rendered by Imaris, at the request of another reviewer) and also cleaned up the dots and lines to better illustrate the aggregate movement of puncta between the first and last frames.

2, Line 238-240, to avoid confusion with Line 243-245, I suggest to expand this sentence, to something like “the number of ribbon puncta remained unchanged from the control (my guess, please check), but the number of postsynaptic puncta was reduced, making the number of paired synapses reduced following KA treatment, consistent with previous studies.”

Thank you for this suggestion. We changed this as suggested:

“The number of ribbon puncta remained unchanged from the control, but the number of post-synaptic puncta was reduced, making the number of paired synapses reduced following KA treatment consistent with previous studies.^{38-40”}

3, Figure 3A, 3B and 3C, please remove “24h” from “24h pre”, unless live imaging was indeed performed 24h before the drug treatment (if this is the case, please say so explicitly in the methods). My understanding is that the control data was obtained not long before the drug treatment, if not immediately before. Also, I would like to suggest to change “Control” to “Vehicle”, and say explicitly what is “Vehicle” (de-ion water, saline, DMEM?) in the methods.

You are correct; the “pre” timepoint was immediately prior to drug treatment, so we removed “24h” as suggested. We used media as the vehicle for the control, and so changed “control” to “media.”

4, Line 275-276, to align better with the other section titles, I suggest to change this title to something like “RIBEYE-tagRFP puncta are more mobile in noise-exposed juvenile, mature-hearing cochlear cultures”.

Thank you for this suggestion. We changed the section title to:

“RIBEYE-tagRFP puncta are more mobile in noise-exposed juvenile, mature-hearing cochlear cultures compared to unexposed counterparts.”

5, Line 306-308, I suggest to delete this sentence, because you are not making a claim about high vs. low spontaneous-rate fibers. If you do, I suggest to choose ribbons in the quarter on the base and toward the outer hair cells.

We removed this sentence as recommended.

6, Line 443, I would like to see the discussion of how an acquisition time of 5 mins could affect the measured velocity of moving ribbons, does it cause underestimation or overestimation?

We added to the Discussion as recommended. Despite the decrease in precision and resolution with the extended acquisition time, we would not expect a systematic over- or underestimation of the speed, since the puncta should be captured at essentially the same point in the z-stack from frame to frame.

“For large movements or rapidly drifting preparations, this might result in a systemic underestimation or overestimation of the punctal speed even after adjusting for the reference frame, if the imaged punctum was captured within a significantly different imaging plane from one frame to another. However, the preparations were quite stable, with minimal movement of the reference frame, and the punctal movement were small; therefore, it is likely that despite the prolonged acquisition time, the measured speeds are likely to be accurate.”

Reviewer #2 (Remarks to the Author):

The study of Mohamad and colleagues addresses an important question of synaptic ribbon dynamics after noise exposure. The authors address this question by using a novel tool, cochlear explant expressing fluorescently-tagged ribbons, of mice subjected to noise exposure

just prior to imaging. The authors first validate the model using neonatal cultures exposed to KA to evoke direct glutamate excitotoxicity and then expand the model to the mature organs from hearing animals. The authors show that both, direct glutamate excitotoxicity as well as noise overexposure induce increase in ribbon volume, unanchoring from synaptic tethers and increased mobility of the ribbons. Upon noise exposure some ribbons seem to move towards nucleus and back, however there seems to be no overall reduction in the number of the ribbons, neither upon KA treatment nor upon noise exposure. I would like to congratulate the authors on performing these technically challenging experiments. This is a very important development of the technique that should allow to address interesting and important questions on the mechanisms of noise noise-evoked synaptopathy in the future. The current limitation of the study and technique is how long the acute preparations can be maintained and imaged and a very poor time resolution. Nevertheless, the MS opens a path for further improvements of the methodology and is thus very important and novel in my view. Furthermore, even with the limited time resolution, it does reveal some interesting consequences of glutamate excitotoxicity and noise overexposure on the presynaptic ribbons. I have a few comments on the present work.

Major:

1. As mentioned above, the main limitation of the study is the limited time resolution. Nevertheless, if I am not mistaken, the fact that the ribbon punctae were tracked only once every 5 minutes is only revealed at the very end of the discussion. Since this knowledge is fairly important for judging the parameter estimations such as speed and velocity, it would be fair to readers to state that clearly somewhere at the beginning of the MS and also to be a bit more conservative in how reliable and valid the assessment of the velocity/speed of puncta is. If the majority of the spots seem to “move” randomly, the “speed” of puncta movements may be significantly underestimated with an imaging limited to every 5 minutes. I understand that this was the current technical limitation of the study as the cells were imaged throughout the whole volume and the method was just recently established for these purposes, still, it would be fair to state this much earlier and add appropriate cautionary statements.

Thank you for this comment. We completely agree that the time resolution is a significant limitation. We intend to perform future experiments imaging highly limited volumetric regions of interest to improve temporal resolution, but agree that for the current manuscript we should be more early and consistent with pointing out this limitation. We have addressed this in the following locations:

In the Methods:

“Acquisition of the entire confocal Z-stack took approximately 5 minutes. For dynamic measures of displacement and speed, a pair of Z-stacks was performed one immediately after the other, thus resulting in an average minimum time interval between images of 5 minutes, which determined the temporal resolution of live timecourse imaging.”

In the Results:

Acquisition of one complete confocal Z-stack required 5 minutes on average; for serial measurement, therefore, the typical minimum time interval between successive images was 5 minutes, which determined the temporal resolution for dynamic measurements.

In the Discussion:Limitations (together with added discussion on imaging limitations requested by another reviewer):

“Adequate 3D imaging of puncta required high-spatial-resolution laser-scanning confocal microscopy across the entire synaptic region, with each z-stack requiring 5 minutes to acquire on average, thus limiting temporal resolution. For large movements or rapidly drifting preparations, this might result in a systemic underestimation or overestimation of the punctal speed even after adjusting for the reference frame, if the imaged punctum was captured within a significantly different imaging plane from one frame to another. However, the preparations were quite stable, with minimal movement of the reference frame, and the punctal movements were small; therefore, it is likely that despite the prolonged acquisition time, the measured speeds are likely to be accurate at the 5-minute time resolution limit. We are unable to make conclusions, however, about movement occurring faster than this limit; future studies restricting imaging to small volumetric regions of interest to enable faster image acquisition are needed to evaluate faster synaptic movement.”

2. The loss of ribbons within 24 hours of vehicle treatment seems quite substantial, judged from the images perhaps 25-30%. The explants were cultured for 24h prior to the first imaging, if I understand correctly. I assume the cultures kept losing the ribbons for at least the next 24 hours, is this degeneration due to culturing? Please, comment in the MS. Is it so that mostly the very small ribbons disappear (when looking at the violin plot in B, the small “balloons” below the large center one seem to disappear or thin out)? Where are these cultures kept between the imaging sessions? Please, specify in the methods.

Thank you for this comment. We too were surprised by the decrease in ribbons after 24h of culture, even in vehicle-treated conditions. This may be due to routine evolution of synapses while in culture (that had been never observed previously without serial imaging of the same specimen) or could be a consequence of some phototoxicity. We added a section to the Discussion:Limitations addressing this baseline decrease in synaptic ribbon number in neonatal cultures:

“We observed other changes that occurred during routine culture and imaging: even in untreated cochleae the number of puncta detected in neonatal cultures decreased after 24h in culture, and the speed of punctal movement decreased over the first 20 minutes of serial imaging, (Figure 3A,E). The reasons for these changes are unknown, but may reflect either a phototoxic response to the initial imaging itself or simply evolution of synapses in explant cultures that had previously been undetectable without serial imaging of the same preparation. These findings illustrate the necessity for tight control of culture period and inclusion of appropriate control conditions in any study using neonatal cochlear cultures, whether using live or fixed imaging techniques.”

We did not see a statistically significant change in the average volume of puncta after 24h culture; while it is true that the small “balloons” corresponding to tiny puncta appear to decrease, these were a tiny fraction of the detected puncta and would not account for the significant decline in punctal number. The cultures were all maintained in the standard tissue-culture incubator in between imaging sessions. This was clarified in the Methods:

“For experiments in which dynamic measurements at multiple timepoints 2-24h apart, cochleae were returned to the CO₂ tissue-culture incubator in between imaging sessions.”

3. What does immediately after KA treatment mean? Immediately after application of KA or immediately after stop of KA treatment (t.i. immediately after 2h KA application)? Based on the speed on Figure 3E, I assume this must mean immediately after 2h KA-application (t.i. immediately after “KA removal”), or? Please, specify. Are those 5 representative timecourses per condition (I assume the authors recorded many more?)?

We have clarified the measurement timepoints relative to KA application. Figures 3A-C refer to an experiment where we performed measurements before KA application (pre) and 24h after KA application (24h post). Figures 3D-E refer to an experiment where we performed serial measurements during KA exposure, starting with a measurement immediately after KA application (Figure 3D) and then for the following 1h during KA application (Figure 3E). Relevant changes:

“We then performed live imaging of RIBEYE-tagRFP neonatal cochleae and obtained serial confocal z-stacks of IHCs before and 24 hr after 2-hr application of 0.4 mM KA.”

And then, next paragraph:

“These speed measurements were performed using a single pair of images before and after 2-hr KA treatment. We then performed serial timecourse imaging during KA exposure. The increase in speed was present immediately after KA exposure (Figure 3D) and remained elevated over a 1-hr recording period (Figure 3E).”

4. From the panels F-G in the Figure 3 it seems like the KA-group does not move more neither in the apical-basal nor in the modiolar-pillar direction. If anything, it is the other way round. So, it seems like the KA-treated ribbons move more, but less in the one or the other given direction, as also the authors point out. But, does that mean that they move more randomly or just along some other trajectories, which are along neither of the two investigated axes? Have the authors looked at the mean square displacement of the puncta over time to identify how much the ribbons travel and in what way?

We did look at the complete 3D mean velocity vector for the puncta (which is, from the standpoint of direction, representative of displacement), and the 95% confidence interval for this measurement crossed the origin; therefore, we interpret this as meaning that the entire population of ribbons, taken as a whole, does not exhibit any significant directionality in movement.

We clarified this in the Results:

“The 95% confidence interval of the mean velocities for overall movement as well as in each of the relevant cochlear axes spanned the origin, suggesting that the entire population of ribbons, taken as a whole, does not exhibit any significant directionality in movement. Indeed, the velocity vectors corresponding to punctal movement along the apical-basal and modiolar-pillar axes were evenly distributed in both control and KA-treated cochleae”

While we did identify the population of basalmost puncta that did exhibit some directionality after noise exposure in juvenile cochlea, we cannot completely exclude that there exist other specific populations of puncta (defined by age, treatment, volume and/or location) that also have specific trajectories of movement.

5. Again, I would like to congratulate the authors on tracking the ribbons in the mature cochleae.

From the quality of the prep to the control of movement and correction for that, the task must have been anything but trivial. It would thus be very nice if the authors explained a bit more in the methods on the technical aspects of the study of how they corrected for the potential movements of the preps. In the acute explants, I can very well imagine that the cells may not be static and possibly also change the shape in the course of the experiment. If that is the case, how was the image/individual cell registered in that case and what was taken as the reference point(s) based on which the ribbon movement was calculated? The authors speak about a stable 3D framework. How stable is that? With other words, while FM1-43 is a great reference, how did the authors account (correct) for the coordinates if the cellular shape changed? Or did that not happen? Also, connected to that, were the organs somehow fixed or did the slide and the coverslip with the customized hole provide good “fixation” of the preps in the chamber? I am not sure if that is explained well enough. And did that actually prevent the issues above, like shape change, large movements etc...? It would be nice if the authors gave some more information on that.

Thank you for this comment. It was indeed challenging to settle on the best way to perform the dissection and mounting, and to establish and adjust for the reference frame. Because of current software limitations with imaris, we are only able to account for the full FM1-43-defined reference frame, and not any cell-specific shape changes. We do demonstrate, in a new **Supplementary Figure S3**, that the FM1-43 reference frame itself is actually quite stable and does not move much from one frame to the next, under both control and KA-/noise-treated conditions. In future studies, we aim to accommodate more precisely for cell shape and cell membrane location using membrane-specific live dyes in more restricted imaging regions.

We added text to the Results to refer to this new Figure:

“The stability of this reference frame was measured in each experimental condition and confirmed to be both stable and not vary with experimental perturbation (Figure S3).”

The juvenile preparations themselves were quite stable (demonstrated in aforementioned new Supplementary Figure S3). We have also created a new **Supplementary Figure S1** illustrating the dissection and mounting technique in more detail. The temporal bone fragment “snaps” into place into the hole, thus providing for very stable fixation on the cover slip. We are currently experimenting with different mounting techniques that may be improve efficiency and have less risk for trauma while maintaining stability.

We added text to the Methods to refer to this new Figure:

“The bone covering the apical turn of the cochlea was flicked off with a #11 scalpel, preserving the membranous labyrinth and exposing the helicotrema. Reissner’s membrane was removed (Figure S1)”

6. The animals were exposed to 8-16kHz noise band at 98 dB SPL for 2 hours, which is a noise paradigm often used in the field. Since there is variability between the results obtained in different labs and different conditions (based on background, age, experimental setups, gender etc), did the authors check the consequences of this exposure on hearing in their animals in these or prior experiments (ref?)? Is it causing the TTS in their animals? Or, did the immunohistochemistry reveal a typical loss of synapses in the midcochlear and basal regions?

7. Interestingly, 2h 8-16kHz 98 dB exposure in 8w-old CBA male mice caused a temporary loss of ribbons in the apical cochleae (Liberman et al., 2015), which recovered within 24h. The data

of the present study does not seem to show that. It would be interesting to see how well these studies compare (e.g. in terms of hearing damage or loss of synapses in a more basal part of the cochlea, see my previous point).

Thank you for these related comments. Similar to the Liberman 2015 study, using this noise exposure paradigm in the RIBEYE-tagRFP mice, we did observe a TTS at mid frequencies with partial TTS at the highest frequencies; in our case, TTS at 8 and 16 kHz, and TTS with partial PTS at 32 kHz. The Liberman data suggest more complete recovery at 32 kHz with the main PTS seen at 45 kHz, which we did not test. Therefore, these hearing results suggest a similar, though not identical, pattern of damage with the noise exposure protocol in our transgenic mice compared with the more thoroughly characterized CBA strain. Mouse strain-specific differences in post-noise-exposure synapse response certainly exist, and so in the future, we plan more thorough characterization of the hearing outcomes in this transgenic line with different noise exposure levels so that we may make comparisons with other published literature.

We have included these ABR data in a new Supplemental Figure S6 and noted the strain-specific differences in synaptopathy in the Discussion:

“Prior reports showed that synaptopathy occurs rapidly during noise exposure, with limited subsequent recovery²⁴, though strain-specific differences have been observed⁵⁴.”

8. What I furthermore find interesting, is, that while the average number of puncta per IHC do not change 2w after exposure, the variance seems to increase a lot. Have authors thought of a potential meaning to that, e.g. loss vs repair and (over-)repair?

We agree that the increase in variance was surprising. In general, the response to noise, even in inbred mice, can be quite variable, both in terms of synaptopathy and threshold shifts. It is certainly possible that this increase in variance is due to differences between animals in terms of their responses to noise, including over/under-repair. Addressing this question is beyond the scope of the current study, but we certainly intend to do a more thorough investigation of ribbons looking at multiple earlier timepoints after synaptopathic noise exposure in a future study.

We have called out this observation in an added sentence in the Results:

“As was observed immediate after noise exposure, the number of pre-synaptic ribbons was not different between noise-exposed and unexposed animals, though variance increased significantly, which may reflect long-term differences between animals in synaptic changes after noise exposure (Figure 5F).”

9. Please, specify the gender of the animals used for noise exposure.

A mix of male and female mice were used for the noise exposure experiments; we have added the sexes used to the revised manuscript (Figure 5 legend). We have not seen any sex-based differences in pilot studies, but a thorough investigation of the effect of sex on ribbon response to noise is beyond the scope of this initial manuscript and will be addressed in a future study.

10. I do not intend to be overly critical, just an observation: The authors suggest that their “basal” puncta based on the orientation should correspond to the ribbons connecting the HSR fibers and the “apical” then, I assume, the LSR and MSR fibers (?). If so, I would expect the basal (HSR) puncta to be smaller rather than larger than the other. Or, to be precise, if anything,

not to see a difference as that is in mice anyway not so large. However, the authors seem to see the opposite. How do you comment on that, a coincidence?

Thank you for pointing this out. Another reviewer pointed out that our distinction between HSR/LSR fibers based on apical/basal location was not entirely precise, so we removed that sentence at their recommendation. (*“Given the orientation...”* Lines 306-308). Therefore, we do not think that the basal/apical volume difference is reflective of HSR/LSR differences. In the future, we are figuring out how to more efficiently and reliably segment out individual hair cells using the FM1-43 signal, which may enable us to more accurately segregate ribbons based on precise cellular location and thereby be more confident in identifying HSR/LSR-associated ribbons.

11. Are the puncta that vanish randomly distributed or do they belong to one or the other group (modiolar-pillar etc.)?

The puncta that “vanish” are randomly distributed. We clarified this in the Results:

“There was no difference in subcellular distribution between puncta that persisted or vanished.”

12. Figure 6F: I do not understand the x-axis of the figure, e.g. how is the velocity at point 0 determined? If I understand correctly, the positions of the puncta were determined every 5 minutes, so the authors actually look at how much the puncta moved from 0 to 300 s, from 300 to 600s etc.. Is that what this image shows? If not, would that not be the most straightforward to show? Also, would it not make more sense to rather display the mean displacement of the puncta rather than velocity?

We acquired our images after delineating a volume of interest that encompassed all of the inner hair cells in the field of view. Because this volume was different from one specimen to another, the time required to acquire the image varied as well. To maximize speed of acquisition, we imaged continually, so that the interval between images was equal to the acquisition time. For this reason, different specimens had different timepoints for each acquired image. To aggregate across specimens to compile this timecourse, we therefore combined images acquired at different timepoints (but within the specified range) relative to $t=0$, and indicated these ranges in the x-axis legend. $T=0$ was always the very first image acquired, so it is precisely $t=0$. The subsequent timepoints did encompass the indicated ranges of times after $t=0$. We added the range of the final timepoint (1200-1800) to address this, and also changed the scale to minutes rather than seconds for space/formatting reasons.

Minor:

1. Methods, line 139, and results line 264. “Time from euthanasia to initiation of imaging averaged 10 minutes.” (results: “...within 10 minutes”) If all given times are correct, is that realistic? I suppose we can assume that loading with dye and three HBSS washes have taken a minute, 5 minutes was reportedly incubation in 37°C, which adds up to 6 minutes. So, the experimenter was left with 4 minutes in total to prep carefully, mount the explant on the slide and under the microscope, and find the perfect position to image under the microscope and initiate a recording. And that on average? Or even maximally (as stated in the results)?

We did time each dissection from euthanasia to imaging and the average was indeed 10 minutes; however, we did not account for the 5-minute incubation time, so the average time was, instead 15 minutes. We corrected this in the Methods and Results. We have also included pictures of the dissection itself in an updated **Supplemental Figure S1**

2. Methods, line 143: “1.1 dwell time”. Time unit?

The unit is microseconds/pixel. This has been added:

“Optical sections in the x–y plane were recorded at 2x averaging, 1.2 AU pinhole, 1.1 microseconds/pixel dwell time”

3. Line 245: Do the authors want to say not affected by glutamate excitotoxicity within 24 hours? Significant losses of ribbons were observed previously after longer times, e.g. a few days after KA exposure, right?

We have clarified this as requested:

“This is consistent with prior findings that the number of pre-synaptic ribbons is not affected by glutamate excitotoxicity within 24 hours^{25,41}”

4. Line 243: correct to 24h after KA “or vehicle” treatment, or something similar

We have corrected this (with some additional clarifications as requested by another reviewer:

“Control cochleae exhibited a decrease in the number of RIBEYE-tagRFP puncta after 24 hr in culture. Cochleae treated with KA also showed a decrease in puncta number; the number of puncta, however, was not statistically significantly different between control and KA-treated cochleae at any timepoint.”

5. Figure 3: I assume all “24h pre” labels should be changed to “pre”. Or were they performed 24h before treatment?

You are correct; the “pre” timepoint was immediately prior to drug treatment, so we removed “24h” for “24h pre” in Figure 3.

6. Figure 4: it might perhaps be wise to add midcochlear and apical to the labels of the panels B-D as that in addition to the age difference may be a contributor to the difference in the number of puncta?

Thank you. We added “midcochlear” and “apical” to the “neonatal” and “juvenile” labels, respectively, to clarify the site of imaging.

Reviewer #3 (Remarks to the Author):

Mohamad and colleagues present a study on the influence of medium noise trauma on inner ear ribbon synapses, as observed using a fluorescent marker. They find that synaptic ribbons appear to become more mobile in organotypic cultures of immature organs of Corti treated with kainic acid as well as in acute explants from noise-treated hearing mice. In the latter, they also find increased size of ribbons after noise trauma, as well as an apparent directional movement of ribbons from the basal pole of the inner hair cells. The increased size and mobility of ribbons persist even two weeks after noise stimulation.

The findings are interesting and well-presented, with sufficient detail to recreate the experiments and adequate analysis, but they do not present a major gain of new information. It has been described before that ribbons appear to move away from the synapse following noise exposure

and that ribbon size increases (e.g. Kujawa and Liberman, J Neurosci 2009; Liberman et al., JARO 2015). As the authors note, this is however the first time that this has been described in living, non-fixed cells. As such, it is a confirmation of existing data using other means, which is also an important contribution, so I support publication.

There are however some parts of the manuscript which are either not clear to me or could be improved:

- I think the conclusions about directional movement of ribbons should be discussed more carefully. The observation is that ribbons from the basal pole of the hair cell have a higher tendency to move towards the apex after noise trauma. However, since the bottom end of the hair cell is restricting downwards or sideways movement (simply because it is a cul-de-sac) detached ribbons are probably more likely to move in an apical direction, simply because this is the only way to go. In this regard the data in Fig. 6F might also indicate that ribbons re-attach after ~20 minutes, so that after moving down they stop moving, which would result in on average increased downward movement.

Thank you for this comment. While we agree that the basal-most ribbons might be expected to only be able to move apically (because the bottom of the hair cell is a cul-de-sac), we found that in control, unexposed animals, the basalmost ribbons actual did NOT show a preference for apical movement. Also, ribbons in neonatal cochleae exposed to KA moved more, but there was no directionality seen even for the basalmost ribbons. It was only after noise exposure in juvenile cochleae that this basal population of ribbons demonstrated apical movement. Therefore, we feel that this likely represents true directional movement of these ribbons induced by noise. We expanded on this in the revised Discussion:

“Compared to all ribbons in control, non-noise exposed animals as well as apical ribbons in noise-exposed animals, which exhibited no directionality of movement, basal ribbons moved towards the apex immediately after noise exposure. While it is possible that these basal ribbons in juvenile, noise-exposed mice are moving apically simply because they have no other direction to go after being unanchored by noise exposure, we did not observe this directionality in basal ribbons in neonatal cochleae after KA exposure, suggesting that this directional movement is specific to noise-exposed mature hearing cochleae. This initial apical movement of the basal-most ribbons towards the hair-cell nucleus was followed by movement back towards the base of the hair cell, which may reflect re-attachment. These findings provide direct visual evidence of dynamic, directional movement of basal synaptic ribbons after noise exposure in the mature hearing cochlea.”

- Following up on the theory that the higher undirected speed of ribbons originates from detached ribbons, it would be interesting to relate the movement speed with the distance of the identified puncta from the hair cell membrane (which was nicely reconstructed, as visible in Fig. 2B).

Thank you for this suggestion. Unfortunately, the FM1-43 signal, especially at the cell periphery, was not sufficiently resolved to precisely delineate the cell membrane and make conclusions based on individual puncta (vs the aggregate locations of puncta across the entire specimen.) We are working on using a live cell membrane dye and perform faster imaging in restricted volumes so that we can follow more precisely individual puncta near the cell membrane.

- It would be interesting to assess the relationship between size of the ribbons and the

integrated fluorescence intensity from the ribbon volume. This would indicate if larger ribbons meant additional RIBEYE material or just less dense (“puffy”) ribbons. The authors remark about a “close correlation between the size and mean brightness of pre-synaptic puncta” (line 379), I am not quite sure what they are referring to here, since I did not find that data in the manuscript.

Thank you for this suggestion. We created a plot of ribbon size vs mean intensity and indeed found that the larger ribbons in juvenile mice were more diffuse and less dense. We added this plot to Figure 4 as a new panel E, and added text as follows:

“Though punctal volume was increased in juvenile cochlea, mean intensity was decreased, suggesting that the puncta become less dense with age (Figure 4E).”

The comment in Line 379 refers to this correlation, and was left as written.

- Why was no imaging performed during the application of KA?

Serial imaging was indeed performed during the application of KA, and is shown in Figure 3E.

- It is stated that KA treatment decreased the number of paired synapses. How were paired synapses identified? Figure S2 does not make it obvious how one could identify them with the large amount of GluR2 spots visible.

Paired synapses were counted by looking at each presynaptic ribbon and counting it as a paired synapse if there is a directly attached GluR2-positive punctum. An example of a paired synapse is shown in a new inset in Figure 1B. Paired synapse counting was done by an investigator blinded to the treatment, so we are confident that the difference we are seeing is correct.

- The authors validate their technique. It would help here if they would also show examples of which puncta were recognized as spots by their Imaris settings and which weren't. Figs. 2A-C give a rough indication, but are not clear.

Thank you for this suggestion. We created and added a new series of panels as Figure 2C demonstrating which puncta are recognized by Imaris and rendered as spheroids.

Minor points:

- It is not quite clear to me whether the experiments on juvenile mice were performed in semi-intact cochleae or organ of Corti explants. I assume the former, but this should be stated more clearly.

We used semi-intact cochleae. We have created a new Supplementary Figure S1 with photographs illustrating the juvenile cochlear dissection.

- Line 86: The authors state that ribbons are not required for establishing synapses in vivo in the retina. The same is true for inner hair cells, as shown by Becker et al. and Jean et al., both eLife 2018. These should be cited.

Thank you. We have added these references to the Introduction:

“Similarly, in RIBEYE knockout mice lacking synaptic ribbons in hair cells, only minor disruptions in the function of hair-cell synapses and minor auditory impairments are present, suggesting the existence of compensatory mechanisms in this model system³⁰⁻³². However, RIBEYE was shown to play a crucial role in organizing presynaptic CaV1.3 calcium channels, although the localization of these channels at the hair-cell synapse remained unaffected. In the absence of RIBEYE, numerous small CaV1.3 clusters were observed at each synapse, deviating from the usual single organized structure.^{30,32}”

- Line 100: “gift of R.L.” should be spelled out.

This has been corrected.

- Line 117/118: What do the abbreviations PBT1 and PBT2 mean?

PBT1 and PBT2 refer to the solutions described in detail in parentheses. We removed these labels for clarity:

“After 3 PBS washes, and blocking (94% PBS, 5% normal goat serum, and 1% Triton X-100, 30 min at RT), cultures were incubated in primary (24 hr, RT) and secondary antibodies (1.5 hr, RT) in 98.6% PBS, 1% normal horse serum, and 0.4% Triton X-100”

- Line 143 “1.1 dwell time” lacks the unit (μs ?). Also, it would be helpful to indicate the frame rate (assuming that it was identical in all experiments).

We added the units for dwell time (microseconds/pixel) and frame rate (0.257 microseconds).

- Line 158 “Auto depth placement was based the FM1-43...” should be “based on the FM1-43”.

This has been corrected.

- Line 210: 75% colocalization, where did this number come from? Is this based on a threshold for pixel intensities?

This colocalization was based on the Pearson’s coefficient, multiplied by 100. We changed “colocalization” to “positive correlation” and report simply the Pearson’s coefficient:

“...a cytofluorogram scatter plot shows a strong positive correlation (Pearson’s coefficient: 0.75) between RIBEYE-tagRFP and anti-Ctbp2 signals.”

- Line 268: “significantly higher (Figure 4C), consistent with prior reports” should better be something like “significantly higher (Figure 4C). Both observations are consistent with prior reports”

Thank you - we have revised as suggested.

- Line 437-439: If the stacks took at least 5 minutes to acquire, how can there be 6 images in a 20 minute recording?

The stacks took 5 minutes on average. For some cochleae, the volume of interest for imaging was smaller and thus the stacks could be acquired faster. We clarified the 5-minute average as follows:

“...with each z-stack requiring 5 minutes to acquire on average, thus limiting temporal resolution.”

- Line 480-481: “top2 and “bottom” should be “left” and “right”

This has been corrected

- Figure 1F: The axes should have units, even if it is only “fluorescence intensity (a.u.)”

Thank you – we have added axis units (fluorescence intensity (a.u.)

- Figure 2B, C: What are the grey boxes in the bottom of the images?

We do not see these gray boxes in our version; it's possible they occurred during file conversion. We have reconfigured and reformatted Figure 2 and hope that these gray boxes are gone.

- Figure 6F: Why are exact times presented for the first and last point, but time ranges for the two in the middle? Movement can only occur between time points, so points 1 and 4 should also indicate a range?

We acquired our images after delineating a volume of interest that encompassed all of the inner hair cells in the field of view. Because this volume was different from one specimen to another, the time required to acquire the image varied as well. To maximize speed of acquisition, we imaged continually, so that the interval between images was equal to the acquisition time. For this reason, different specimens had different timepoints for each acquired image. To aggregate across specimens to compile this timecourse, we therefore combined images acquired at different timepoints (but within the specified range) relative to $t=0$, and indicated these ranges in the x-axis legend. $T=0$ was always the very first image acquired, so it is precisely $t=0$. The subsequent timepoints did encompass the indicated ranges of times after $t=0$. We added the range of the final timepoint (1200-1800) to address this, and also changed the scale to minutes rather than seconds for space/formatting reasons.

REVIEWERS' COMMENTS:

Reviewer #1 (Remarks to the Author):

All my previous comments have been addressed with changes in figures and text, and I have no further comments.

Reviewer #2 (Remarks to the Author):

The authors have addressed my concerns. Small cosmetics: the axis labels including the numbers look awkward in some figure panels (either too small or bit misplaced). I suggest to carefully check all figures. Fig. 3E x-axis could benefit from omitting some time labels.

Reviewer #3 (Remarks to the Author):

Mohamad and colleagues have followed up on my suggestions and I believe their manuscript to be improved by this. I have only minor comments:

- I still don't understand why a relationship between ribbon speed and distance from membrane can't be made (see response to my second comment from the first round of reviews). While I understand that the FM1-43 labeling won't allow a properly resolved labeling of the membrane, it is nonetheless used in the manuscript to determine the outlines of the cell. Did the authors try to relate the movement speed with the distance from the membrane and did not find any significant effect or did they simply think that this is not a promising approach? This is not a critical point, however, but simply mentioned out of interest.

- The response to my question "Why was no imaging performed during the application of KA?": I now understand that the imaging was done during exposure and not after. It would have been better to include one data point before application of KA into the graph, since this would allow for better comparison with the untreated data. If such data exist, I would appreciate if they could be added, if not then it also works as is.

Otherwise, I'm happy with the manuscript as is and would very much support publication in Communications Biology

Thank you very much for your careful review of our revised manuscript. We have addressed the remaining comments below.

REVIEWERS' COMMENTS:

Reviewer #1 (Remarks to the Author):

All my previous comments have been addressed with changes in figures and text, and I have no further comments.

Reviewer #2 (Remarks to the Author):

The authors have addressed my concerns. Small cosmetics: the axis labels including the numbers look awkward in some figure panels (either too small or bit misplaced). I suggest to carefully check all figures. Fig. 3E x-axis could benefit from omitting some time labels.

Thank you – we have reviewed the axis labels and adjusted appropriately, including removing excess time labels in Fig 3e.

Reviewer #3 (Remarks to the Author):

Mohamad and colleagues have followed up on my suggestions and I believe their manuscript to be improved by this. I have only minor comments:
- I still don't understand why a relationship between ribbon speed and distance from membrane can't be made (see response to my second comment from the first round of reviews). While I understand that the FM1-43 labeling won't allow a properly resolved labeling of the membrane, it is nonetheless used in the manuscript to determine the outlines of the cell. Did the authors try to relate the movement speed with the distance from the membrane and did not find any significant effect or did they simply think that this is not a promising approach? This is not a critical point, however, but simply mentioned out of interest.

Thank you – we agree that looking for associations between distance from membrane and ribbon speed is very important. We did attempt to conduct this analysis using our current model, by measuring the distance between each punctum and the closest point on the rendered surface as defined by the FM1-43 signal. Upon this preliminary analysis, we did not see any association between distance from the membrane and ribbon speed in either control or noise-exposed cochleae. However, we do have concerns about frame-over-frame variance in the exact surface rendering, which compromises our confidence in this membrane-ribbon distance measurement. That is, although the overall FM1-43 signal is

adequate for drift correction for the entire 3D volume (in a way that enables us to stabilize the gross imaging frame and make us confident that we are tracking puncta from one frame to another without systematic movement artifact), we feel like the FM1-43-defined surface is not adequately precise to accurately determine ribbon distance from the membrane. We are developing a method to use live membrane dyes and much more restricted imaging volumes to enable greater temporal and spatial precision, which should allow us to address this question properly in a future study.

- The response to my question "Why was no imaging performed during the application of KA?": I now understand that the imaging was done during exposure and not after. It would have been better to include one data point before application of KA into the graph, since this would allow for better comparison with the untreated data. If such data exist, I would appreciate if they could be added, if not then it also works as is. Otherwise, I'm happy with the manuscript as is and would very much support publication in Communications Biology

Unfortunately, while we did capture a single pre-drug timepoint in the timecourses presented in Figure 3e, we did not capture the two timepoints necessary to calculate pre-drug speed for comparison between media- and KA-treated cochleae.